# Yin Yang 1 sustains biosynthetic demands during brain development in a stage-specific manner

Luis Zurkirchen[1], Sandra Varum[1], Sonja Giger[1], Annika Klug[1], Jessica Häusel[1], Raphaël Bossart[1], Martina Zemke[1], Claudio Cantù[2,5], Zeynep Kalender Atak[3], Nicola Zamboni[4], Konrad Basler[2] & Lukas Sommer[1]

The transcription factor Yin Yang 1 (YY1) plays an important role in human disease. It is often overexpressed in cancers and mutations can lead to a congenital haploinsufficiency syndrome characterized by craniofacial dysmorphisms and neurological dysfunctions, consistent with a role in brain development. Here, we show that Yy1 controls murine cerebral cortex development in a stage-dependent manner. By regulating a wide range of metabolic pathways and protein translation, Yy1 maintains proliferation and survival of neural progenitor cells (NPCs) at early stages of brain development. Despite its constitutive expression, however, the dependence on Yy1 declines over the course of corticogenesis. This is associated with decreasing importance of processes controlled by Yy1 during development, as reflected by diminished protein synthesis rates at later developmental stages. Thus, our study unravels a novel role for Yy1 as a stage-dependent regulator of brain development and shows that biosynthetic demands of NPCs dynamically change throughout development.

[1] Institute of Anatomy, University of Zurich, 8057 Zurich, Switzerland. [2] Institute of Molecular Life Sciences, University of Zurich, Zurich 8057, Switzerland. [3] Laboratory of Computational Biology, KU Leuven Center for Human Genetics, Leuven 3000, Belgium. [4] Institute of Molecular Systems Biology, ETH Zurich, Zurich 8093, Switzerland. [5] Present address: Wallenberg Centre for Molecular Medicine (WCMM), Department of Clinical and Experimental Medicine (IKE), Linköping University, Linköping 58183, Sweden. Correspondence and requests for materials should be addressed to L.S. (email: lukas.sommer@anatomy.uzh.ch)

Y Y1 syndrome is a rare intellectual disability disorder, which is caused by mutations in the human *Yin Yang 1* (*YY1*) gene[1]. Deletions and point mutations of *YY1* lead to a congenital haploinsufficiency syndrome characterized mainly by cognitive impairment, facial dysmorphisms, and developmental delay. Interestingly, in mice, a subset of embryos lacking one *Yy1* allele (*Yy1*[+/−] heterozygous) exhibit exencephaly, pseudoventricles, and brain asymmetry[2]. Although this has not yet been experimentally addressed, the combined data are consistent with the hypothesis that Yy1 has a role in mammalian brain development.

The gene product of *Yy1* is a ubiquitously expressed transcription factor, which controls transcriptional activation and repression and has been implicated in enabling enhancer–promoter interactions[3,4]. Yy1 exhibits context-dependent roles during the development and homeostasis of many tissues. It has been shown to regulate muscle[5,6], lung[7,8], and cardiac development[9] and intestinal stem cell development and homeostasis[10,11]. Despite its ubiquitous expression, Yy1 seems to regulate distinct steps during the development of these tissues. Depending on the cell type, Yy1 has been associated with various functions, including regulation of signaling molecules, survival signals, cell cycle regulators or metabolism[5,7,11–13]. In the brain, a recent study using short hairpin RNA (shRNA) against Yy1 suggested a role for Yy1 in promoting neural progenitor cell (NPC) differentiation at mid-neurogenesis[14]. Likewise, Yy1 has been shown to be required for proper differentiation of the oligodendrocytic lineage at postnatal stages in vivo[15]. Although it is still unclear how cell type-specific functions of such an ubiquitous factor are achieved, the central nervous system and craniofacial structures appear to be especially dependent on the activity of YY1 as evidenced by the phenotype of YY1 loss-of-function in human patients[1].

In this report, we genetically ablated *Yy1* specifically in the developing dorsal cortex of mice. Loss of Yy1 before the onset of neurogenesis resulted in microcephaly owing to the depletion of NPCs. We found that ablation of *Yy1* induced transient $G_1/S$ phase cell cycle arrest and p53-dependent cell death at embryonic day 12.5 (E12.5). In contrast, deletion of *Yy1* after the onset of neurogenesis demonstrated a continuously decreasing influence on proliferation and cell survival. At the molecular level, loss of Yy1 at early developmental stages impaired numerous biosynthetic pathways, notably influencing the expression of metabolic genes, metabolite abundance, and protein translation rate. Intriguingly, at later stages of cortex development, Yy1 inactivation did not affect metabolic processes anymore and the rate of protein synthesis was generally reduced in later stage NPCs, revealing stage-dependent demands for metabolism and protein translation in cortical development.

## Results

### Yy1 regulates NPC survival and proliferation

To study the role of Yy1 in cortex development, we started by determining the expression pattern of Yy1 at various developmental stages. Quantitative real-time polymerase chain reaction (QRT-PCR) analysis and immunostaining demonstrated that Yy1 mRNA and protein were prominently expressed throughout cortical development, with a slight decrease in overall expression levels at late developmental stages (Supplementary Fig. 1a–c). Notably, Yy1 protein was detectable in virtually all Sox2+ NPCs and doublecortin (Dcx) + neuronal cells at all stages analyzed (Supplementary Fig. 1c). To address the in vivo requirements of Yy1 in the developing cortex, we conditionally ablated *Yy1* by combining *Emx1-Cre* mice[16] with a transgenic mouse line carrying *Yy1* alleles flanked by *loxP* sites (*Yy1*[lx/lx] mice[17]) (Fig. 1a). In *Emx1-Cre Yy1*[lx/lx] mice (hereafter, referred to as *Yy1cKO*), Yy1

protein expression was lost in single cells as early as embryonic day 10.5 (E10.5) and was completely absent in the dorsal cortex from E11.5 onwards (Supplementary Fig. 1d).

At E18.5, the forebrain of *Yy1cKO* mice was considerably smaller than that of control mice, which was reflected by significantly reduced cortical thickness and number of cells per radial unit (RU = 100 μm) (Fig. 1b and Supplementary Fig. 1e–g). Importantly, mesenchymal craniofacial structures known to influence forebrain development[18,19] and to be malformed in human YY1 syndrome[1] were not affected, consistent with the known Emx1 expression pattern[16]. Throughout our analysis, *Emx1-Cre Yy1*[lx/wt] mice appeared morphologically normal and were viable. This is consistent with previous reports showing lack of haploinsufficiency in mice for a number of congenital diseases, which indicates differential sensitivity towards gene dosage in humans vs. rodents[20,21]. To investigate the cellular mechanisms underlying the reduction of cortical size in *Yy1cKO* mice, we analyzed a potential requirement of Yy1 for proliferation at early stages of corticogenesis (Fig. 1c, e and Supplementary Fig. 1h–k). Immunohistochemical analysis revealed reduced numbers of cells positive for the mitotic marker phosphorylated histone H3 (pHH3) in *Yy1cKO* brains at E12.5 as compared with the control (Fig. 1c, e), whereas the ratio of apical vs. basal pHH3+ was not changed (Fig. 1f). Likewise, at E12.5, *Yy1cKO* cortices displayed strongly decreased numbers of cells expressing CyclinD1, which is required for cells to progress through the G1 phase of the cell cycle (Fig. 1g, i). In contrast, expression of CyclinB1, important for the transition from $G_2$ to M phase, was not changed upon conditional *Yy1* ablation (Fig. 1j, l).

To confirm the requirement of Yy1 for cell cycle progression, we isolated NPCs from E11.5 cortices by microdissection to perform siRNA-mediated knockdown (KD) experiments (Fig. 1p–r and Supplementary Fig. 1o). CyclinD1 expression was decreased in siYy1 KD-cells, whereas CyclinB1 expression remained unchanged (Fig. 1q). Moreover, cell cycle profile analysis by flow cytometry showed that treatment with siYy1 RNAs retained the cells preferentially in G1, indicating a role of Yy1 in G1 phase progression (Fig. 1r and Supplementary Fig. 1p).

Apart from proliferation, survival of cortical cells was strongly affected upon *Yy1* inactivation. Indeed, immunostaining for cleaved Caspase 3 (cCasp3) revealed transiently increased apoptosis in the forebrain of *Yy1cKO* mice at E12.5 (Fig. 1m, o). Similarly, the number of Annexin V + cells was significantly higher in siYy1-treated cortical cells than in control cells (Fig. 1s), demonstrating a role of Yy1 in promoting cell survival. In contrast, Yy1 is apparently not involved in neuronal differentiation as such. Although at E18.5 the numbers of neuronal subtypes normally found in distinct cortical layers were generally decreased upon *Yy1* inactivation, consistent with the reduced cortical thickness at this stage, the relative abundance of Dcx + immature neurons in comparison with Sox2-expressing NPCs was similar in *Yy1cKO* and control forebrains at E12.5 (Supplementary Fig. 2a–g).

Thus, our data suggest that Yy1 controls proper cortex size by regulating proliferation and survival of cortical progenitor cells. Intriguingly, however, when analyzed at E15.5 rather than at E12.5, the amount of cortex cells expressing pHH3, CyclinD1, and cleaved Caspase 3, respectively, was not affected anymore in *Yy1cKO* embryos (Fig. 1d, e, f, h, i, k, l, n, o), pointing to a specific requirement of Yy1 at early stages of development.

### Stage-specific control of NPC behavior by Yy1

To assess potential stage-specific roles of Yy1 in cortical development, we made use of the *CreER*[T2] *loxP* system, by which activation of CreER[T2] recombinase can be induced at defined stages of

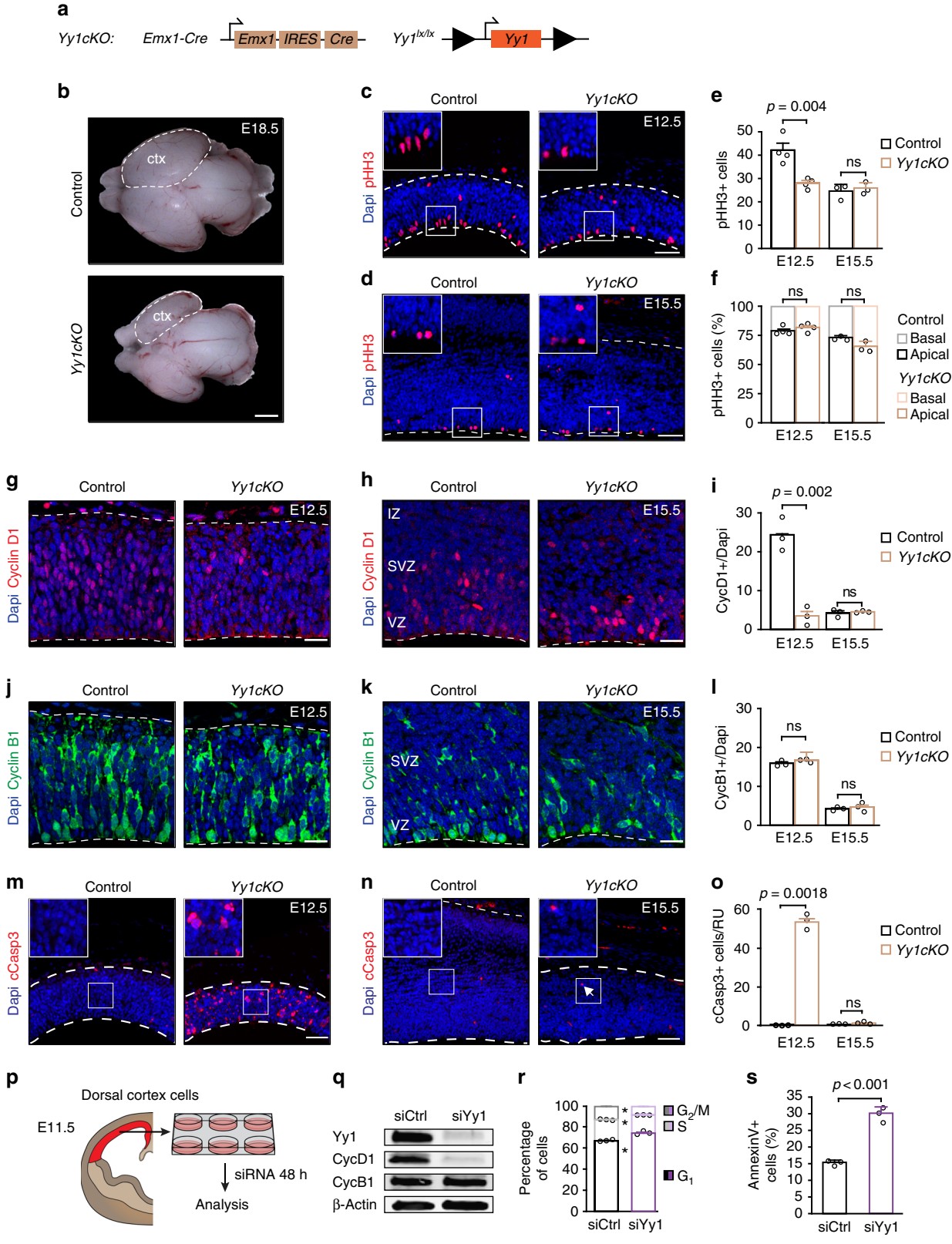

embryonic development upon tamoxifen (TM) injection of pregnant females. To this end, $Yy1^{lx/lx}$ mice were mated with $Emx\text{-}CreER^{T2}$ $Yy1^{lx/wt}$ mice ($Emx\text{-}CreER^{T2}$ from[22]) to obtain animals suitable for inducible deletion of $Yy1$ in the developing cortex (hereafter referred to as $Yy1iKO$ mice) (Supplementary Fig. 3a). Non-inducible $Emx1\text{-}Cre$ is known to be active in the developing cortex from around E9.5 onwards[16]. Accordingly, TM-induced activation of $Emx\text{-}CreER^{T2}$ in $Yy1iKO$ mice at E9.5 temporarily decreased pHH3+ cells and CyclinD1 expression and increased apoptosis at E12.5 (Supplementary Fig. 3b–l), in analogy to the phenotype described above for $Emx1\text{-}Cre$-carrying $Yy1cKO$ mice (Fig. 1). Interestingly, however, when $Yy1$ deletion

**Fig. 1** Yy1 maintains proliferation and cell survival at early stages of cortex development. **a** Genotype of *Yy1cKO* mice with conditional ablation of *Yy1* in the dorsal cortex. **b** Deletion of *Yy1* leads to decreased cortex (ctx) size at E18.5. **c–f** Loss of Yy1 decreases the number of pHH3+ cells at E12.5 (**c**, **e**). At E15.5, the number of mitotic cells is comparable to control embryos (**d**, **e**). The ratio of apical vs. basal pHH3+ cells does not change upon knockout of *Yy1* (**f**). The number of pHH3+ cells is normalized to 600 μm ventricular zone length (E12.5 and E15.5) and normalized to cortical thickness (E15.5). **g–i** The percentage of CyclinD1+ cells decreases upon ablation of *Yy1* at E12.5 (**g**, **i**), but not at E15.5 (**h**, **i**). **j–l** The percentage of CyclinB1+ cells is not affected in *Yy1cKO* embryos. **m–o** Immunohistochemistry for cleaved Caspase 3 (cCasp3) shows that the number of apoptotic cells transiently increases at E12.5 in *Yy1cKO* embryos. Radial unit (RU) = 100 μm. **p** Experimental outline for knockdown experiments. **q** Efficient knockdown of Yy1 in isolated E11.5 cortical progenitor cells decreases CyclinD1 protein levels without affecting CyclinB1 protein levels. **r** Cell cycle analysis by flow cytometry reveals that knockdown of Yy1 increases the number of cells in G1 cell cycle phase. **s** Flow cytometric analysis of the apoptotic marker Annexin V. Knockdown of Yy1 in isolated E11.5 cortical progenitor cells increases the number of Annexin V+ cells. Nuclei are counterstained with DAPI. Scale bars represent 1 mm (**b**), 100 μm (**m**, **n**), 50 μm (**c**, **d**), 20 μm (**g**, **h**, **j**, **k**). Comparisons were performed using the two-tailed unpaired Student's *t* test. Data are the mean ± standard deviation. *$p < 0.05$. ns = not significant

was induced at E10.5 or later, the effect on forebrain size was progressively decreasing when compared to *Yy1cKO* mice, despite efficient ablation of Yy1 protein expression (Fig. 2a–c and Supplementary Fig. 3m–r). In agreement with this, ablation of *Yy1* at E12.5 affected proliferation and cell death less prominently at E14.5 and both of these parameters were at control levels at E15.5 (Fig. 2d–k). Moreover, upon TM-induced recombination at E13.5, the size of *Yy1iKO* cortices at E18.5 was indistinguishable from that of control embryos (Fig. 2b, c), and neither cortical cell proliferation nor apoptosis were significantly altered at E15.5 in these *Yy1iKO* embryos as compared with the control (Fig. 2l–q). Thus, despite constitutive expression of Yy1 at later stages of cortical development (Supplementary Fig. 1c), NPC proliferation and survival are becoming less dependent on Yy1 as the embryonic brain develops, revealing a stage-dependent role of Yy1.

With advancing cortical development, the relative number of intermediate progenitor cells compared to apical NPCs increases[23]. To assess whether the phenotype seen in *Yy1* mutant embryos can be attributed to different progenitor cell types, we performed immunohistochemistry for Pax6 and Tbr2 to distinguish apical NPCs (Pax6+), intermediate NPCs (Tbr2+), and apical NPCs transitioning to intermediate NPCs (Pax6+ Tbr2+). At E12.5 and E15.5, both the number of Pax6+ and Tbr2+ cells per RU were decreased in *Yy1cKO* embryos (Supplementary Fig. 4a, b, c, e, f). The relative abundance of these NPC subtypes, however, were not altered (Supplementary Fig. 4d, g). Consistent with our previous results, TM-induced ablation of *Yy1* at E12.5 showed a milder phenotype where only the number of Pax6+ cells per RU at E14.5 was significantly decreased (Supplementary Fig 4h–o). Further, deletion of Yy1 at E13.5 did neither affect the absolute nor relative number of Pax6, Tbr2, and Pax6+ Tbr2+ cells (Supplementary Fig. 4p–s). Together, our data suggest that loss of Yy1 exerts a stage-specific effect on cell proliferation and survival in the developing brain, which cannot just be attributed to different sensitivities of distinct NPC subpopulations.

**Yy1 promotes NPC survival through downregulation of p53.** Yy1 has been shown to inhibit activation of the tumor-suppressor p53, a well-established regulator of cell cycle progression and apoptosis[24–26]. To address whether Yy1 might regulate p53 also in the developing CNS, we investigated p53 expression in the forebrain of control and *Yy1cKO* mice at E12.5. Consistent with the reported posttranscriptional regulation of p53 by Yy1[24,25], p53 protein but not p53 mRNA levels were upregulated upon loss of Yy1 (Fig. 3a, b). To address a potential involvement of p53 in Yy1-dependent neural progenitor control, we then crossed *Yy1cKO* mice with mice carrying alleles of *Trp53* flanked by *loxP* sites[27] (Fig. 3c). The resulting *Yy1 Trp53* double knockout (*Yy1Trp53dKO*) mice displayed a cortex with a size intermediate

to control and *Yy1cKO* cortices at E18.5, indicating that loss of p53 could partially rescue the *Yy1cKO* phenotype (Fig. 3d, e). Of note, immunostaining for cleaved Caspase 3 demonstrated that the increased cell death observed upon *Yy1* inactivation (Fig. 1m, o) was fully rescued in the absence of p53 (Fig. 3f). In contrast, loss of p53 did not rescue the proliferation phenotype seen in *Yy1cKO* mice, as shown by quantification of pHH3 and CyclinD1 expression (Fig. 3g–k).

To substantiate these data, we inactivated p53 signaling in *Yy1cKO* mice by administration of the pharmacological inhibitor Pifithrin-α (PFTα) (Supplementary Fig. 5a)[28,29]. Again, this approach counteracted cell death induced by conditional *Yy1* deletion, albeit not as efficiently as genetic p53 inactivation (Supplementary Fig. 5b). However, the proliferation defect in the developing forebrain of *Yy1cKO* mice was still present in PFTα-treated animals (Supplementary Fig. 5c–g). Thus, regulation of cortical progenitor survival by Yy1 appears to be largely mediated by the activity of p53, whereas the control of proliferation by Yy1 is p53-independent.

**Yy1 regulates metabolic pathways and protein translation.** To identify further factors potentially mediating Yy1 activity in the developing forebrain, we evaluated the global gene expression patterns of control vs. *Yy1cKO* cortices at E11.5 using RNA-sequencing (RNA-seq) (Fig. 4a). We identified 1087 and 467 annotated mRNAs that were downregulated and upregulated, respectively, in *Yy1cKO* cells (|log2 fold change| > 0.32, $p < 0.05$, FDR < 0.01). To find which pathways were specifically enriched, we performed gene ontology (GO) analysis and assembled a GO perturbation network, where GO terms, which share > 50% of the genes, are connected (Fig. 4b). GO terms enriched upon deletion of Yy1 revealed alterations in various metabolic pathways, such as fatty-acid oxidation, nucleic-acid metabolism, and biosynthetic processes (Fig. 4b, Supplementary Fig. 6 and Supplementary Data 1). Furthermore, mitochondria were identified as organelles primarily associated with Yy1-dependent changes in gene expression, in agreement with previous studies on muscle development and intestinal stem cell homeostasis[5,11].

To validate these findings, we performed qRT-PCR analysis of selected genes and confirmed that a plethora of factors involved in metabolic pathways including glycolysis, TCA cycle, lipid metabolism, and mitochondrial biogenesis were downregulated in the *Yy1cKO* forebrain (Fig. 4c, d). Moreover, the expression of several genes implicated in nucleotide metabolism, ribosome and tRNA synthesis, and mRNA to protein translation was altered upon *Yy1* inactivation (Fig. 4e). Of note, genes associated with metabolism and translation, were not rescued by pharmacological inhibition of p53 signaling in *Yy1cKO* cortical cells at E12.5 (Supplementary Fig. 7a–c), indicating that Yy1-mediated regulation of these genes is p53 independent in the developing cortex.

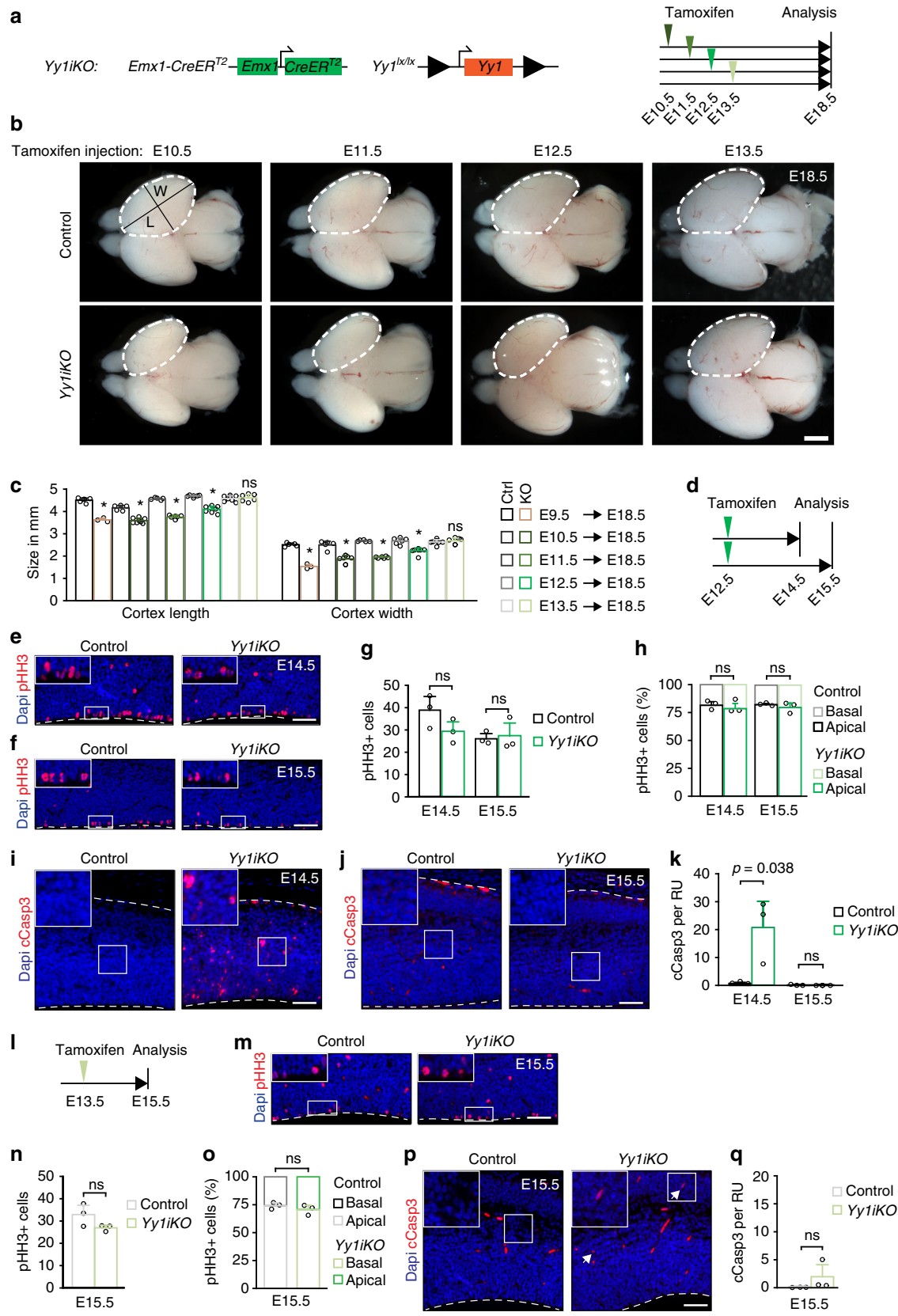

We next analyzed the direct target genes of Yy1 by interrogating genome-wide occupancy of Yy1 using chromatin immunoprecipitation followed by sequencing (ChIP-seq) of cortical cells isolated from E12.5 wild-type embryos (Fig. 5a). The highest scoring motif corresponded to a known Yy1

consensus binding site containing the core sequence ATGGC (Fig. 5b)[6,11,30]. Consistent with previous reports[6,11,30], Yy1-binding events predominantly took place within close proximity of the transcriptional start site (TSS) of promoter regions (Fig. 5c, d). Visualizing GO terms enriched for the genes with Yy1-binding

**Fig. 2** Ablation of *Yy1* after E12.5 does not influence cortical development. **a** Genotype of mice and experimental strategy used to induce ablation of *Yy1* at different developmental stages. **b** Later stage tamoxifen-induced ablation of *Yy1* ameliorates the decrease in cortex size compared with *Yy1cKO* cortices (Fig. 1b). **c** Measurement of cortical length (L) and width (W) as indicated in **b**. For representative picture of E18.5 Yy1cKO cortex, see Fig. 1b. *$p < 0.05$. **d** Experimental strategy to ablate *Yy1* at E12.5 in *Yy1iKO* embryos (for Fig. 2e–k). **e–h** Immunostaining and quantification for pHH3+ cells at E14.5 (**e**, **g**) and E15.5 (**f**, **g**) in E12.5-ablated *Yy1iKO* embryos. **h** depicts the ratio of apical vs. basal pHH3+ cells. The number of pHH3+ cells is normalized to 600 μm ventricular zone length. **i–k** Ablation of *Yy1* at E12.5 elicits cleavage of Caspase 3 (cCasp3) at E14.5 (**i**, **k**) but not at E15.5 (**j**, **k**). Note that red signals visible at E15.5 are blood cells. **l** Experimental strategy to ablate *Yy1* at E13.5 in *Yy1iKO* embryos (for Fig. 2m–q). **m–o** The total number and ratio of apical vs. basal pHH3+ cells remains unchanged upon late ablation of *Yy1* at E13.5. The number of pHH3+ cells is normalized to 600 μm ventricular zone length. **p**, **q** Ablation of Yy1 at E13.5 only induces cell death in a minority of cells (indicated by arrows, remaining red signals are blood cells). Nuclei are counterstained with DAPI. Scale bars represent 1 mm (**b**), 50 μm (**e**, **f**, **i**, **j**, **m**, **p**). Comparisons were performed using ANOVA (Tukey's multiple comparisons test) (**c**) and two-tailed unpaired Student's *t* test (**g**, **h**, **k**, **n**, **o**, **q**). Data are the mean ± standard deviation. ns = not significant

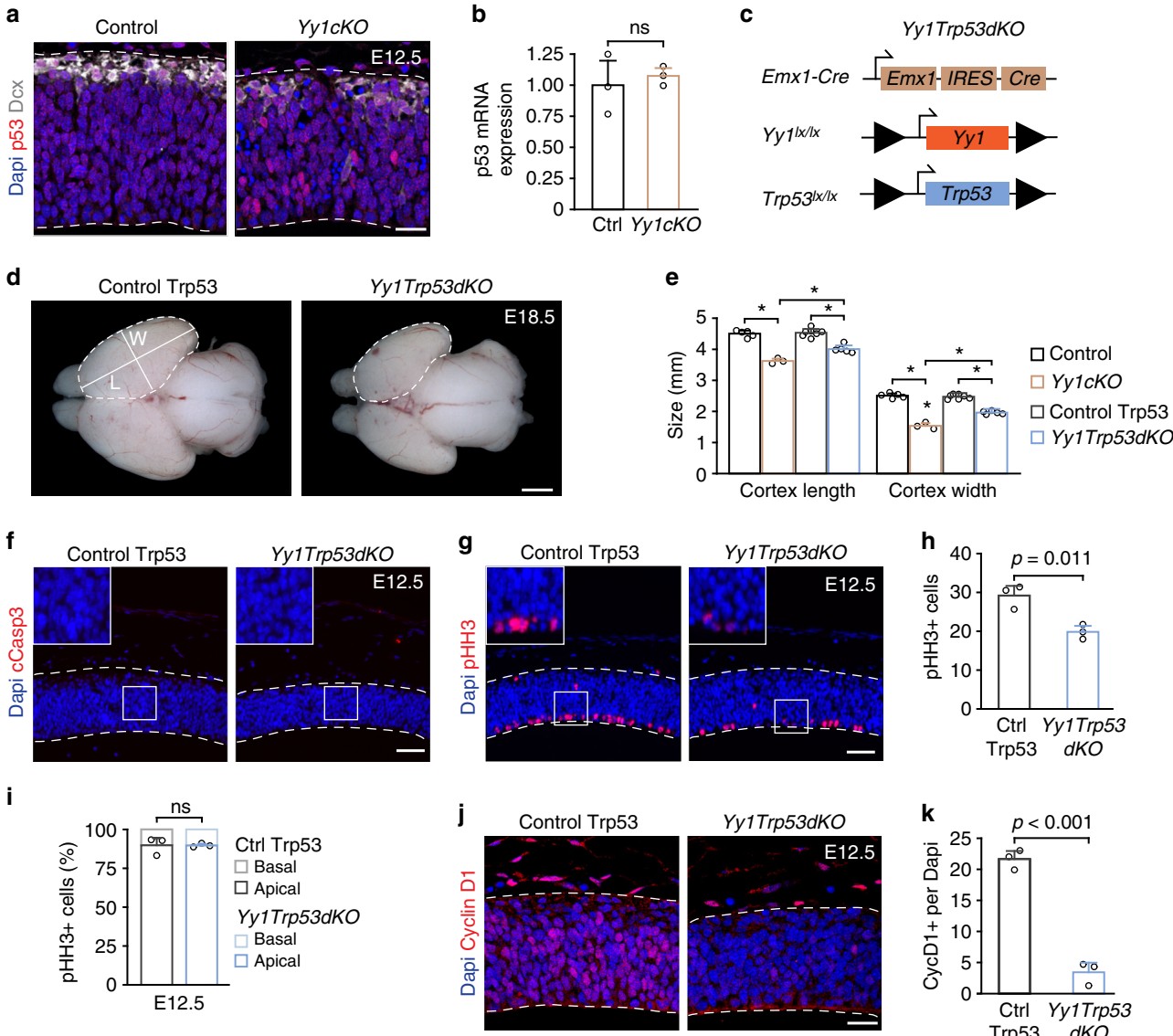

**Fig. 3** Loss of Yy1 induces p53-dependent apoptosis. **a** Accumulation of p53 protein in *Yy1cKO* embryos at E12.5. **b** qRT-PCR of p53 mRNA levels reveals no difference upon loss of Yy1. **c** Genotype of *Yy1 Trp53* double knockout (*Yy1Trp53dKO*). **d**, **e** Double knockout of *Yy1* and *Trp53* partially rescues cortical size at E18.5. Cortical length (L) and width (W) measurements from Control and *Yy1cKO* from Fig. 2c were reused for illustrative reasons. Representative images of control and *Yy1cKO* cortices are shown in Fig. 1b. *$p < 0.01$. **f** Knockout of *Trp53* in the context of *Yy1* ablation completely abolishes emergence of cleaved Caspase 3+ cells. **g–i** *Trp53 Yy1* double knockout does not restore the number of mitotic pHH3+ cells. The ratio of apical vs basal pHH3+ cell does not change (**i**). The numbers of pHH3+ cells are normalized to 600 μm ventricular zone length. **j**, **k** Loss of *Trp53* does not rescue the number of CyclinD1+ cells in *Yy1* mutant embryos. Nuclei are counterstained with DAPI. Scale bars resemble 1 mm (**d**), 50 μm (**f**, **g**), 20 μm (**a**, **j**). Comparisons were performed using the two-tailed unpaired Student's *t* test (**b**, **h**, **i**, **k**) and ANOVA (Tukey's multiple comparisons test) (**e**). Data are the mean ± standard deviation. ns = not significant

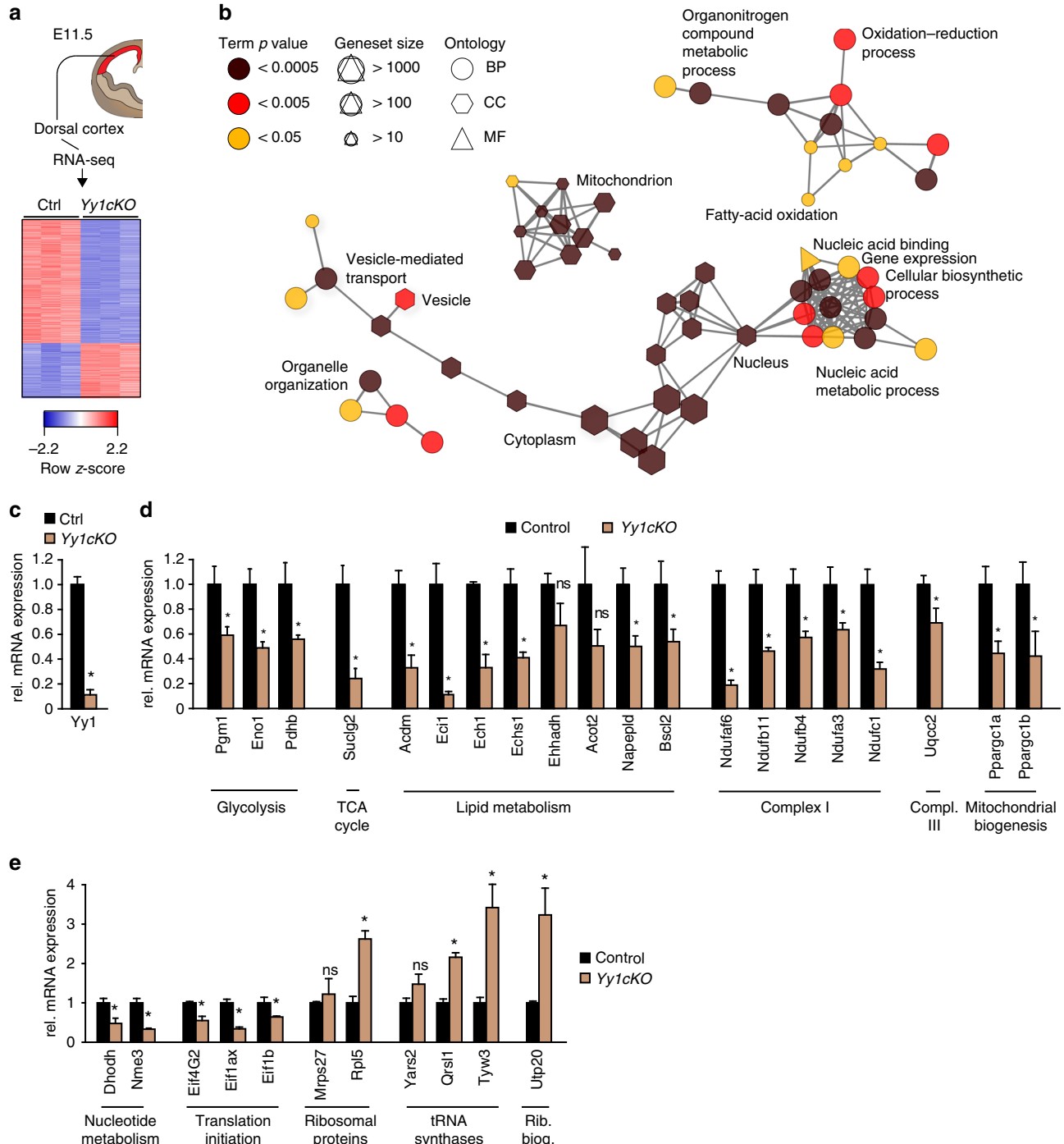

**Fig. 4** Yy1 regulates the expression of metabolic genes. **a** RNA-seq comparing *Yy1cKO* versus control cortex tissue at E11.5 identified 1554 differentially expressed genes (|log2 fold change| > 0.3, *p* < 0.05, FDR < 0.01). **b** Gene Ontology (GO) term network analysis on the basis of differentially regulated genes. Each node represents an enriched GO term (adjusted *p* value (Corrected with Bonferroni step down procedure) < 0.05). Nodes are interconnected when the gene overlap is > 50%, based on the kappa score. BP, biological process; MF, molecular function; CC, cellular component. A fully labeled version of the network is given in Supplementary Fig. 6. **c–e** qRT-PCR validation for differentially regulated genes comparing control vs *Yy1cKO* cortical tissue at E12.5 confirms RNA-seq results. Comparisons were performed using the two-tailed unpaired Student's *t* test. Data are the mean ± standard deviation

motifs in a functionally grouped network demonstrated an association of Yy1-target genes with metabolism and mitochondrial functions as well as with RNA processing, ribosome biogenesis, and protein translation (Fig. 5e–g, Supplementary Fig. 8 and Supplementary Data 2). Thus, Yy1 directly controls expression of numerous gene sets implicated in metabolic and translational processes.

To functionally test whether the impaired expression of nuclear encoded mitochondrial genes affects mitochondrial bioenergetics, we assessed mitochondrial function by measuring the oxygen consumption rate (OCR) of control and *Yy1cKO* cells directly isolated from E12.5 cortices. Real-time measurements of OCR demonstrated that *Yy1* inactivation led to impaired basal respiration, ATP-linked OCR, and maximal respiration capacities

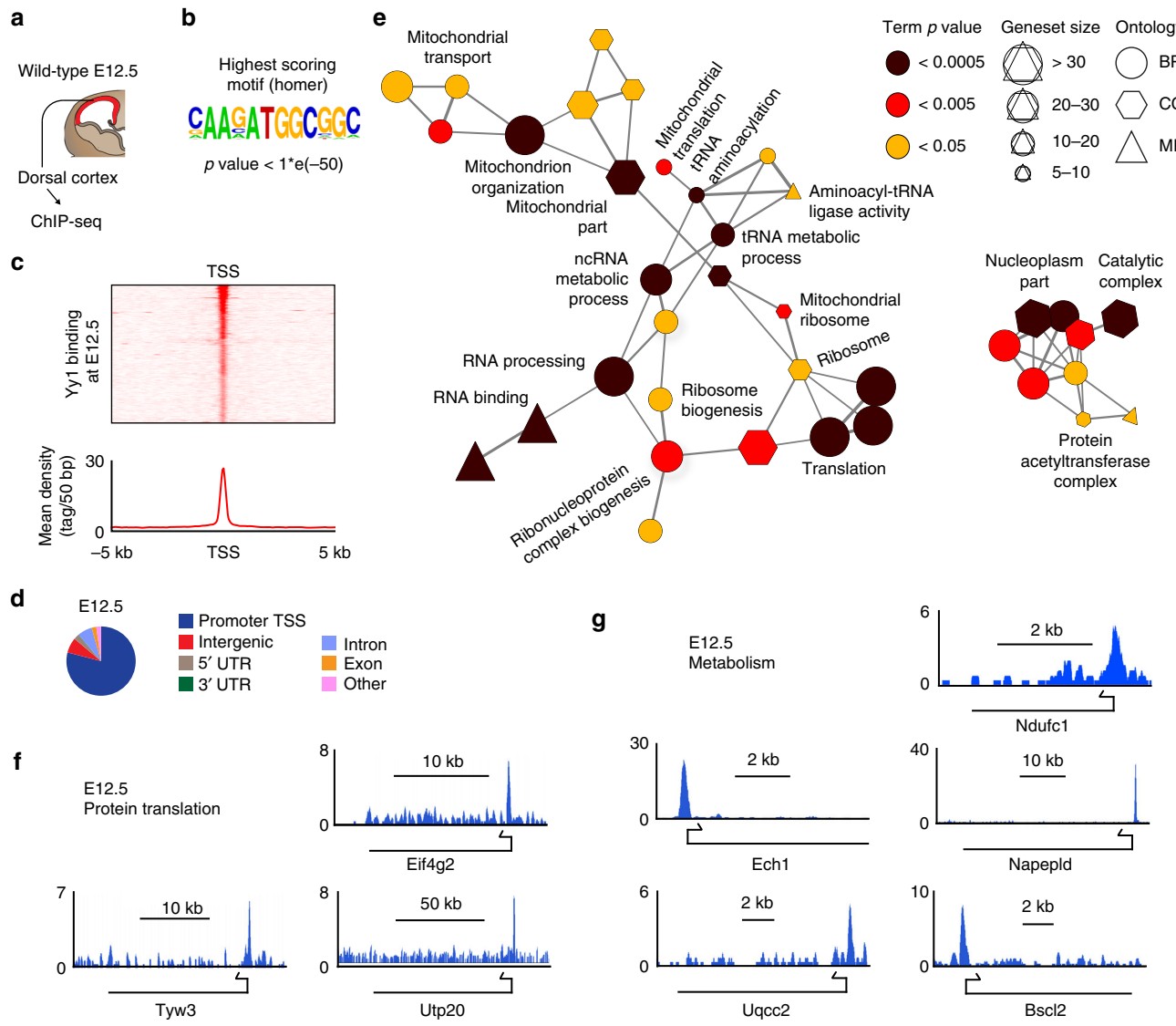

**Fig. 5** Yy1 directly binds to metabolic genes. **a** Genome-wide binding of Yy1 to DNA regions was analyzed by chromatin immunoprecipitation against Yy1 followed by sequencing (ChIP-seq) of cortex cells derived at E12.5. Statistical analysis of the enriched regions identified 464 binding events at E12.5 (FDR < 0.001), present in at least 1 of 2 replicas. **b** Homer motif discovery revealed a known Yy1-binding motif as the highest scoring motif ($p < 1\ast$e-50). **c** Read cluster profile reveals preferential binding of Yy1 close to the transcription start site (TSS) of target genes. **d** Distribution of binding sites to different genomic locations. Yy1 binds preferentially to the promoter region of the transcription start site (TSS) of genes. **e** Gene Ontology (GO) term network analysis on the basis of Yy1-bound target genes at E12.5. Each node represents an enriched GO term (adjusted p value (Corrected with Bonferroni step down procedure) < 0.05). Nodes are interconnected when the gene overlap is > 50%, based on the kappa score. BP, biological process; MF, molecular function; CC, cellular component. A fully labeled version of the network is given in Supplementary Fig. 8. **f**, **g** Genomic snapshots depicting Yy1-binding events at metabolic genes (**g**) and genes involved in protein translation (**f**). kb, kilo bases

of cortical cells (Fig. 6a, b). These data could not simply be explained by reduced amounts of mitochondria, as the ratio of mitochondrial over nuclear DNA (Mit1 and CytB vs intergenic region) was not altered (Fig. 6c and Supplementary Fig 9a). Likewise, we excluded that apoptosis contributed to this phenotype since the number of living cells (7AAD-negative) just before OCR measurement was not altered in control vs. *Yy1cKO* cells (Supplementary Fig 9b, c). To obtain a global overview of metabolic changes induced by loss of Yy1, we analyzed mass spectrometry-based metabolic profiles of control and siYy1 KD cortical cells derived from E11.5 embryos. Quantitative examination of the untargeted metabolomics data showed downregulation of various metabolic processes including glucose, lipid, amino acid, and nucleotide metabolism upon Yy1 KD (Fig. 6d and Supplementary Data 3).

Apart from general metabolic processes, the analysis of our RNA-seq and ChIP-seq data indicated an involvement of Yy1 in protein translation. To functionally validate these findings, we quantified global protein synthesis in cortical cells with either undisturbed or reduced Yy1 levels by measuring incorporation of *O*-propargyl-puromycin (OP-puro) into nascent proteins[31]. In cells isolated from E11.5 cortices and treated with siYy1 RNA, the relative protein synthesis rate was highly reduced as compared with control cells (Fig. 6e, f). Likewise, cells directly isolated from the dorsal cortex of *Yy1cKO* embryos at E12.5 displayed significantly lower translation rates than measured in control cortical cells (Fig. 6i, j). To elucidate whether differences in protein translation might depend on the cell cycle stage, we performed flow cytometric analysis of OP-puro incorporation intensity in combination with propidium

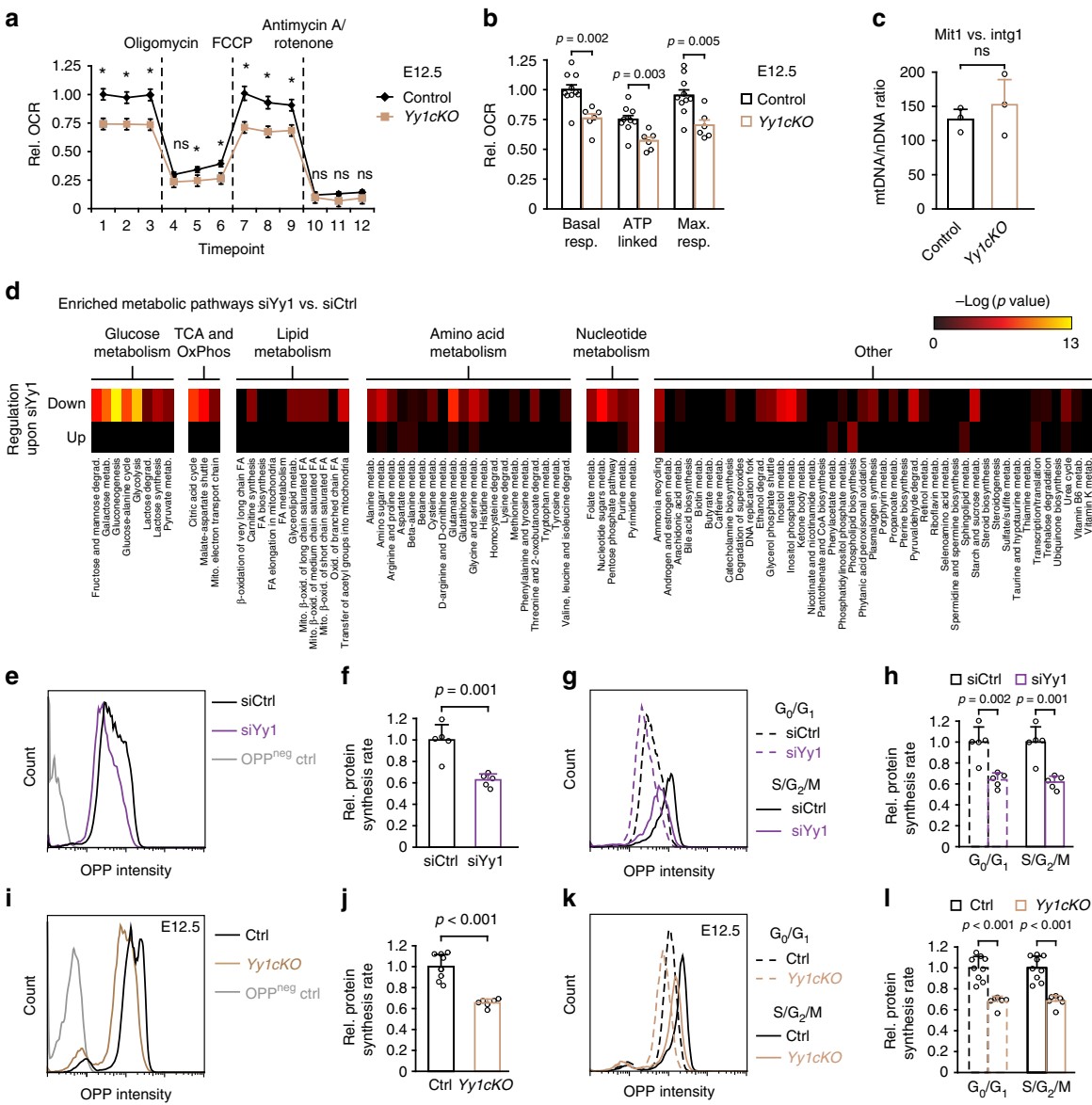

**Fig. 6** Yy1 controls cortical metabolism and protein translation rate. **a**, **b** Oxygen consumption rate (OCR) measurement of isolated cortical cells at E12.5 using a Seahorse Extracellular Flux Analyzer reveals impaired mitochondrial bioenergetics upon ablation of Yy1. Injection of electron transport chain inhibitors are indicated after measurement 3 (oligomycin, ATP synthase inhibitor), 6 (FCCP, mitochondrial uncoupler) and 9 (Antimycin A/rotenone, complex III & I inhibitors). Parameters derived from **a** are indicated in **b**: basal respiration, ATP-linked OCR, and maximum respiration capacity. Data represented relative to first basal respiration measurement of controls and as a mean of $n = 9$ (control), $n = 6$ (Yy1cKO) error bars indicate standard error of the mean. **c** qRT-PCR for mitochondrial DNA content shows no difference between Yy1cKO and control cortex tissue. Graphs present mitochondrial (Mit1) versus nuclear (intergenic region, intg1) DNA ratio. **d** Metabolomic alterations in isolated E11.5 NPCs upon knockdown of Yy1 for 48 h. Heatmap shows enrichment of metabolic pathways which are downregulated or upregulated upon treatment with siYy1. Abbreviations: oxidative phosphorylation (OxPhos), β-oxidation (β-Ox), fatty acid (FA), metabolism (metab.), mitochondrial (mito.), degradation (degrad.). **e**–**h** Knockdown of Yy1 reduces protein translation rate. OP-puro (OPP) intensity histogram of representative siRNA-treated samples pulsed with OPP for 30 min and OPP-negative control (**e**). Quantification of mean fluorescent OPP intensity (**f**). OPP incorporation in siRNA-treated cortex cells in $G_0G_1$ (DNA content $= 2c$) and $S/G_2/M$ (DNA content $> 2c$) phases of the cell cycle (**g**, **h**). DNA content was determined using propidium iodide. **i**–**l** Reduced protein translation rate in Yy1cKO cells at E12.5. OP-puro (OPP) intensity histogram of representative E12.5 control and Yy1cKO cells pulsed with OPP for 30 min and OPP-negative control (**i**). Quantification of mean fluorescent OPP intensity (**j**). OPP incorporation in cortical cells in $G_0G_1$ (DNA content $= 2c$) and $S/G_2/M$ (DNA content $> 2c$) phases of the cell cycle (**k**, **l**). DNA content was determined using propidium iodide. Comparisons were performed using the two-tailed unpaired Student's $t$ test. Data are the mean ± standard deviation (**c**, **f**, **h**, **j**, **l**) and ± standard error of the mean (**a**, **b**). *$p < 0.05$. ns = not significant

iodide-based measurements of DNA content in siYy1 and Yy1cKO cells and their respective control cell populations (Supplementary Fig. 9d). These experiments demonstrated that proper translation was dependent on Yy1 both in $G_0/G_1$ cells (DNA content $= 2c$), which comprise NPCs in G1 phase of the cell cycle and postmitotic neurons potentially present in the cell preparations, as well as in cycling progenitor cells in $S/G_2/M$ phase (DNA content $> 2c$) (Fig. 6g, h, k, l).

**Stage-dependent requirements for biosynthesis in NPCs.** Our study shows that Yy1 controls a broad spectrum of metabolic pathways together with global protein synthesis at early stages of

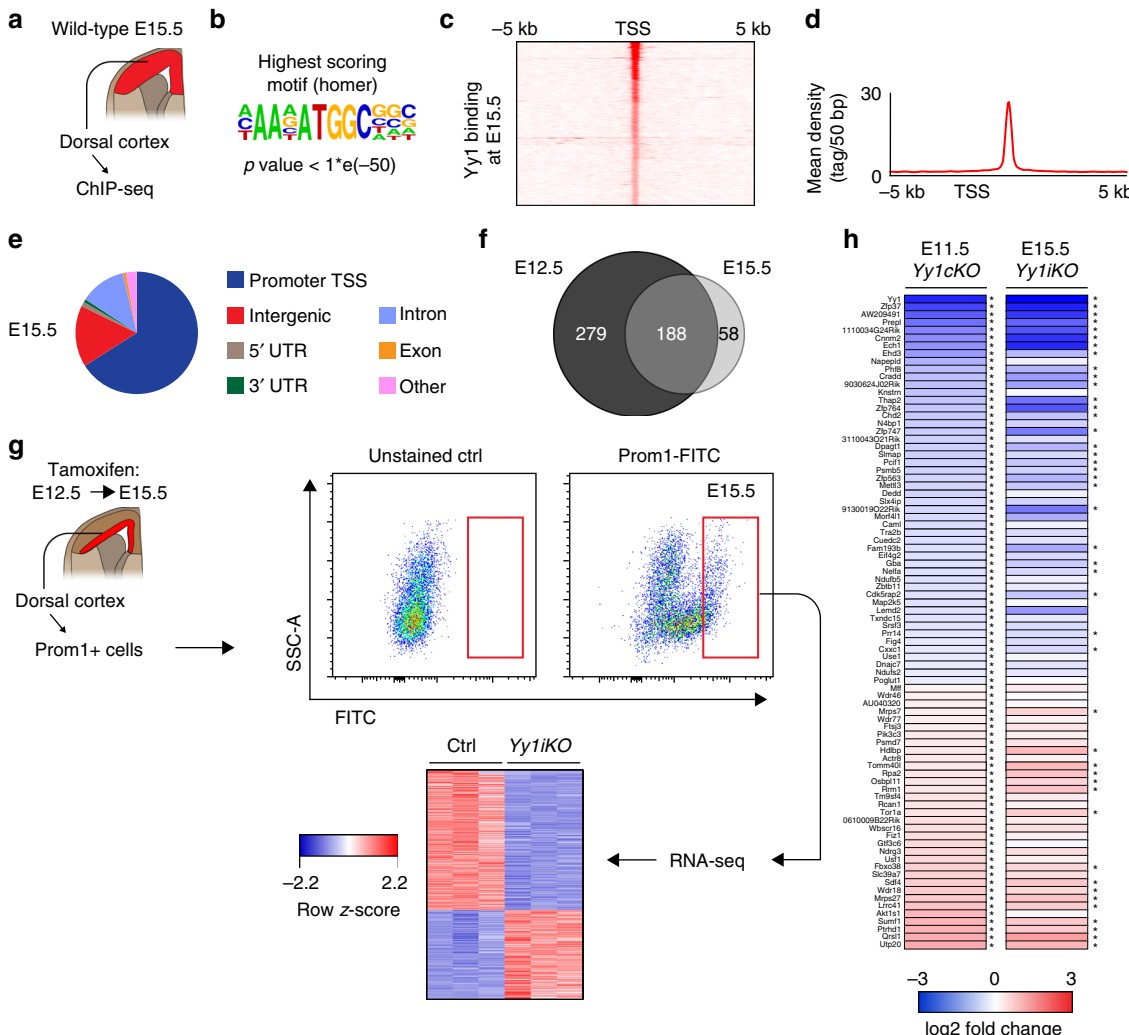

**Fig. 7** Yy1 controls a similar gene set throughout cortex development. **a** Genome-wide binding of Yy1 to DNA regions was analyzed by chromatin immunoprecipitation followed by deep sequencing (ChIP-seq) of cortex cells derived at E15.5. Statistical analysis of the enriched regions identified 246 binding events at E15.5 (FDR < 0.001), present in at least 1 of 2 replicas. **b** Homer motif discovery revealed the known Yy1-binding motif as the highest scoring motif ($p < 1*e-50$). **c–e** Read cluster profile and distribution of binding sites to different genomic locations at E15.5. Yy1 binds preferentially to the promoter region of the transcription start site (TSS) of genes. **f** Analysis of overlap between E12.5 and E15.5 ChIP-seq data. **g** RNA-seq comparing E15.5 *Yy1iKO* vs control (each $n = 3$ embryos) radial glia cells from the dorsal cortex which were isolated by microdissection and subsequent Prominin1-based (CD133-FITC) FACS. A total of 1080 genes were downregulated (660 genes) or upregulated (420 genes) upon induced knockout of *Yy1* (|log2 fold change| > 0.32, $p < 0.05$, FDR < 0.05). **h** RNA-seq expression of genes at E11.5 and E15.5, which are bound by at E12.5 and E15.5, upon knockout of Yy1. *$p < 0.05$

cortical development. The stage-dependent phenotype observed in *Yy1cKO* and *Yy1iKO* embryos (Figs. 1 and 2) raises the question of whether Yy1 functions might change during the course of cortical development. To address this we performed ChIP-seq of wild-type cortical cells at E15.5 (Fig. 7a). Similar to what we observed at E12.5 (Fig. 5), ChIP-seq for Yy1 at E15.5 showed that a Yy1-binding element was the most significantly enriched motif and that most ChIP-seq peaks were found in proximity to the TSS of the target genes (Fig. 7b–e). Surprisingly, binding events which were detected at least once in both ChIP-seq replicas for E12.5 and E15.5, overlapped to a large extent between the two developmental stages (Fig. 7f, Supplementary Fig. 9e, f), indicating that Yy1 exhibits a similar binding pattern at these stages.

Next, we performed RNA-seq analysis of FACS-isolated Prominin1+ radial glia (RG) cells of *Yy1iKO* vs control embryos at E15.5 upon late ablation of *Yy1* (Fig. 7g). A total of 1080 genes were downregulated (660 genes) or upregulated (420 genes) upon

induced knockout of *Yy1* (|log2 fold change| > 0.32, $p < 0.05$, FDR < 0.05). We then compared changes in gene expression of direct Yy1 targets at early and late stages. Although some genes with significantly changed expression ($p < 0.05$, FDR < 0.05) at E11.5 were not changed at E15.5, the majority of genes were similarly deregulated at both early and late stages (Fig. 7h). Likewise, when considering both indirect and direct Yy1 targets, the majority of Yy1-dependent gene products localizing to mitochondria or involved in metabolism and protein translation were similarly deregulated at E11.5 and E15.5 (Supplementary Fig. 10a–e). Accordingly, metabolic processes and lipid metabolism were enriched among GO terms for downregulated genes in *Yy1iKO* embryos at E15.5 (Supplementary Fig. 10f), comparable to the situation at E11.5 (Fig. 4b). In agreement with the observation that Yy1 seems to regulate similar genes throughout cortical development, computational prediction of transcription factor binding by i-cisTarget[32] to find Yy1 co-factors revealed that both at E12.5 and E15.5, the highest enriched motifs were from E2f

and Ets transcription factor family members (Supplementary Fig. 11a, b). Of note, both E2f and Ets family members have previously emerged as important metabolic regulators[33–37].

However, although many genes of metabolic and biosynthetic pathways seemed to be affected by Yy1 at both developmental stages, a small but significant fraction of such genes were differentially regulated at E11.5 but not at E15.5. These included Ndufc1, Ndufb4, Ndufb5 (subunits of the electron transport chain), Ppargc1α (negative regulator of glycolysis, promotes mitochondrial gene expression), Napepld, Cpt1c (lipid metabolism), Eif1b, Rps13, Rpl5, and Akt1s1 (protein translation and ribosome biogenesis) (Fig. 7h and Supplementary Fig. 10a–e). Thus, it is possible that this latter group of genes contributes to the stage-dependent phenotype observed upon ablation of Yy1. In addition, some genes involved in cell cycle progression were upregulated specifically at late stages. However, based on our ChIP-seq analysis, all but two of these genes were not bound and only indirectly regulated by Yy1, pointing to compensatory mechanisms active in E15.5 Yy1iKO NPCs. Moreover, the two bound genes, Cdk5rap2 and Cdc45, were regulated in the same direction at both developmental stages. In summary, our data suggest that the gene regulatory network dependent on Yy1 does not change overtly along development, raising the question of whether later stage cortical cells can cope better with the changes in gene expression induced by Yy1 inactivation.

To functionally address this point, we assessed whether loss of Yy1 after the onset of neurogenesis influences mitochondrial bioenergetics. Intriguingly, basal respiration, ATP-linked OCR, and maximal respiration capacity at E15.5 were not impaired upon late Yy1 inactivation (Fig. 8a–c), unlike at E12.5 after early Yy1 deletion (Fig. 6a, b). Likewise, global protein synthesis rates were only mildly affected at E15.5, especially when taking into account proliferating progenitor cells in $S/G_2/M$ phase (Fig. 8d–g). Importantly, global protein synthesis rates of Prom1+ RG cells of Yy1iKO and control cells were indistinguishable (Fig. 8h, i). Thus, consistent with the decreasing disturbance of proliferation and survival observed when Yy1 was ablated at later developmental stages (Fig. 2), mitochondrial bioenergetics and protein translation were not affected upon Yy1 deletion after the onset of neurogenesis.

Our data are compatible with the hypothesis that the dependency of cortical cells on Yy1 is decreasing at later stages of development, because the main processes regulated by Yy1 might have lost relative importance over time. To address this hypothesis, we measured overall protein synthesis rates at E10.5, E12.5 and E15.5 in wild-type embryos. Levels of OP-puro incorporation were highly reduced in E15.5 cortical cells as compared with E10.5 and E12.5 cells (Fig. 8j, k and Supplementary Fig. 12a, b). Translation was also significantly lower in late than in early-stage cortical cells, when assessing OP-puro incorporation in $G_0/G_1$ and cycling $S/G_2/M$ cells (Fig. 8l, m and Supplementary Fig. 12c, d). Finally, the significant decrease in protein translation rate was also apparent when comparing Prom1+ RG cells at E15.5 vs E12.5 (Fig. 8n, o). Thus, the stage-dependent phenotype in survival and proliferation of NPCs seen upon Yy1 inactivation appears to reflect stage-dependent requirements for biosynthetic processes.

## Discussion
Regulation of NPC proliferation and survival is essential to ensure correct brain development. Aberrations in cell cycle progression and survival of NPCs lead to decreased numbers of NPCs and lie at the heart of many neurodevelopmental disorders[38–41]. In this study, we report a previously unappreciated role for the transcription factor Yy1 in the regulation of brain size. We show that Yy1 is crucial during early cortical development to sustain NPC proliferation and survival, in contrast to developmental stages after the first waves of neurogenesis when the activity of Yy1 seems to be dispensable for cortex development. By using a combination of genome-wide DNA-binding site identification, transcriptomics, metabolomics, and analysis of protein translation rates we demonstrate that Yy1 is a key factor safeguarding biosynthetic demands specific for early-stage NPCs. Our study not only reveals that Yy1 controls the expression of metabolic genes, metabolite abundance and mitochondrial bioenergetics during corticogenesis, but also shows that Yy1 is functionally involved in the transcriptional regulation of global protein synthesis. Possibly, aberrations in these processes may contribute to the neurodevelopmental defects in humans with heterozygous YY1 loss-of-function mutations[1], although this needs to be addressed in future studies.

Conditional ablation of Yy1 at early stages of cortical development resulted in microcephaly owing to decreased cell proliferation and increased cell death. Emx1-Cre- and Emx1-CreER[T2]-dependent recombination leads to ablation of the floxed gene in NPCs as well as neurons[16,22]. However, the observed phenotype appears to be owing to a requirement for Yy1 in NPCs rather than secondary to defects in neurons, given the very low number of neurons present at stages displaying the most prominent phenotype and, vice versa, the absence of a proliferation phenotype at later stages, when many neurons are present. Moreover, neuronal differentiation as such was not affected by the loss of Yy1. In addition, siRNA-mediated knockdown experiments in isolated NPCs recapitulated the Yy1cKO phenotype, suggesting a cell-autonomous effect specifically in NPCs.

Increased apoptosis observed upon ablation of Yy1 was accompanied by accumulation of p53 protein, a tumor-suppressor, which integrates various stress signals and activates downstream effectors to promote the appropriate cellular response: either damage repair and cell survival or induction of apoptosis in irreparable cases[42,43]. Apoptosis can be induced by p53-dependent or -independent processes[44]. By using both genetic ablation and pharmacological inhibition of p53, we show that apoptosis induced by the loss of Yy1 is dependent on p53 and contributes to the microcephaly phenotype. Yy1 has been implicated in the direct regulation of p53 and the phenotype upon loss of Yy1 in thymocytes can be completely rescued by p53 deficiency[26]. In contrast, inhibition of p53 in Yy1cKO embryos neither ameliorated the proliferation phenotype, nor rescued expression of Yy1-dependent metabolic genes. Thus, apart from p53-mediated cell death, loss of Yy1 in the cortex induces a p53-independent stress response that is associated with impaired proliferation of NPCs. Surprisingly, a recent study using shRNA-mediated knockdown of Yy1 suggested that Yy1 limits NPC expansion and promotes differentiation via repression of Sox2 at E14.5[14]. However, our approach to genetically delete Yy1 did neither reveal alterations of Sox2 protein expression nor regulation of Sox2 mRNA levels and did not increase proliferation. Although the reasons for these contradicting findings remain unclear, it could be explained by the efficiency in reducing Yy1 levels using shRNA compared with genetic deletion.

Similar to our findings in the embryonic brain, Yy1 exerts a pro-proliferative role in many types of cancer[45,46], in lung epithelial cells[7] and during cardiac development[9]. Despite the decreased number of CyclinD1-expressing cells and decrease in CyclinD1 mRNA expression in the Yy1cKO cortex, we did not detect direct binding of Yy1 to the promoters of cell cycle progression genes by ChIP-Seq. This suggests that Yy1 does not directly regulate the cell cycle machinery of NPCs, but rather

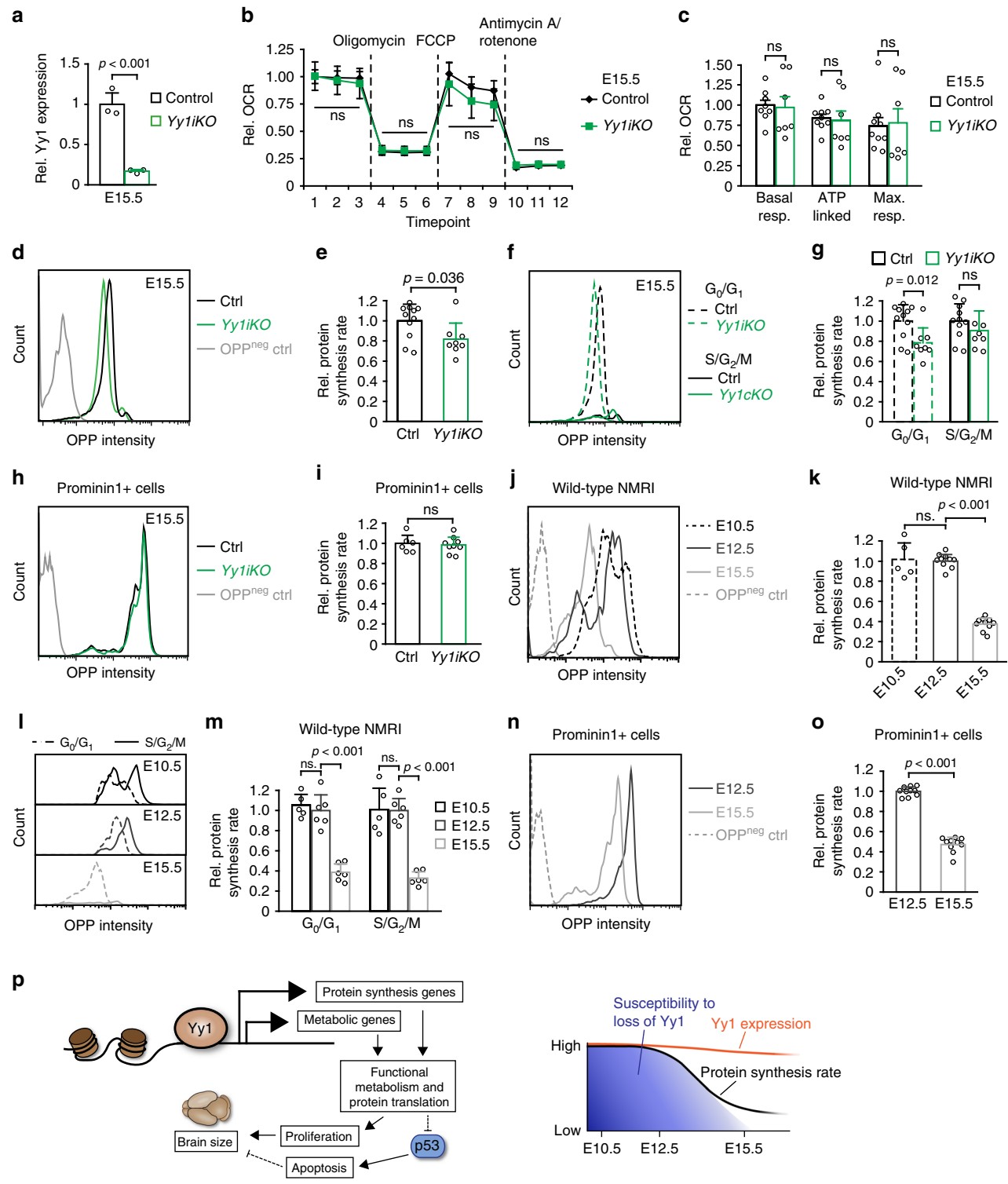

influences other cellular pathways important for proper proliferation of early NPCs. Indeed, our molecular analysis demonstrates that Yy1 is a key transcription factor controlling the expression of genes involved in central metabolic pathways, such as glycolysis, TCA cycle, lipid metabolism, nucleotide synthesis, and subunits of the electron transport chain. In addition, we observed decreased mitochondrial function in absence of Yy1, which can further influence the abundance of precursor molecules for biosynthesis[47,48]. Yy1 has previously been associated

with the regulation of metabolic functions, thereby supporting the development and homeostasis of muscle cells and intestinal stem cells[5,10,11]. Going beyond these previous reports, our RNA-seq and ChIP-seq data sets revealed a very broad implication of Yy1 in various metabolic pathways and also identified several genes involved in ribosome biogenesis, tRNA synthesis, and protein translation to be direct targets of Yy1. Although binding of Yy1 to genes of the protein translation machinery has been observed before[1,6,11,30], this finding has not been explored further. We now

**Fig. 8** Stage-dependent requirements for biosynthesis in NPCs. **a** Efficient reduction of Yy1 mRNA at E15.5 upon tamoxifen-induced (TM) ablation at E12.5. **b**, **c** Oxygen consumption rate (OCR) measurement of isolated cortical cells at E15.5 using a Seahorse Extracellular Flux Analyzer reveals that mitochondrial bioenergetics are not altered upon ablation of *Yy1* at E12.5. Injection of electron transport chain inhibitors are indicated after measurement 3 (oligomycin, ATP synthase inhibitor), 6 (FCCP, mitochondrial uncoupler), and 9 (Antimycin A/rotenone, complex III & I inhibitors). Parameters derived from **b** are indicated in **c**: basal respiration, ATP-linked OCR and maximum respiration capacity. Data represented relative to first basal respiration measurement of controls and as mean of $n = 9$ (control), $n = 7$ (*Yy1iKO*), error bars indicate standard error of the mean. **d**–**g** Protein translation rate is only mildly affected in *Yy1iKO* cells at E15.5 upon recombination at E12.5. OP-puro (OPP) intensity histogram of representative E15.5 control and *Yy1iKO* cells pulsed with OPP for 30 min and OPP negative control (**d**). Quantification of mean fluorescent OPP intensity (**e**). OPP incorporation in cortical cells in $G_0G_1$ (DNA content = 2c) and $S/G_2/M$ (DNA content > 2c) phases of the cell cycle (**f**, **g**). DNA content was determined using propidium iodide. Note that the protein translation rate of cycling progenitor cells is not significantly altered between control and *Yy1iKO* embryos. **h**, **i** TM-induced ablation of *Yy1* at E12.5 does not alter OPP incorporation in Prominin1+ RG cells at E15.5. **j**–**o** Reduced protein translation rate of E15.5 vs E12.5 and E10.5 wild-type cortical cells. OP-puro (OPP) intensity histogram of representative E10.5, E12.5, and E15.5 cells pulsed with OPP for 30 min and OPP-negative control (**j**). Quantification of mean fluorescent OPP intensity (**k**). OPP incorporation in cortical cells in $G_0G_1$ (DNA content = 2c) and $S/G_2/M$ (DNA content > 2c) phases of the cell cycle (**l**, **m**). DNA content was determined using propidium iodide. The reduction of OPP incorporation is also apparent in Prominin1+ RG cells of E15.5 vs E12.5 (**n**, **o**). Comparisons were performed using the two-tailed unpaired Student's *t* test. Data are the mean ± standard deviation (**a**, **e**, **g**, **i**, **k**, **m**, **o**) and ± standard error of the mean (**b**, **c**). ns = not significant. **p** Schematic drawing summarizing the results. The transcription factor Yy1 binds and regulates genes involved in metabolism and protein synthesis. Functional biosynthesis sustains proliferation and survival of NPCs and eventually leads to correct brain development. Despite constitutive expression of Yy1, cortical NPCs exhibit a stage-dependent susceptibility to loss of Yy1, which coincides with decreased protein translation rates at later developmental stages

demonstrate that Yy1 regulates the expression of these genes and thereby functionally controls global protein synthesis rates.

Historically viewed as mere housekeeping functions, metabolism and protein translation have arisen as key players coordinating stem cell behavior. Metabolic processes have recently emerged as crucial factors controlling tissue development and homeostasis, in particular, of the nervous system[49–53]. During cortex development, NPCs have been shown to rely on high glycolytic rates in order to proliferate[50] and to modulate their mitochondrial morphology upon differentiation[51]. In addition, upregulation of electron transport chain components and of oxidative phosphorylation is critical for fast proliferating intermediate progenitor cells during adult neurogenesis[53]. Although little is known about the importance of protein synthesis during brain development, adult hematopoietic[54], hair follicle,[55] and neural stem cells[56] have been demonstrated to synthesize less protein compared with their direct, cycling progeny. Fast proliferating cells therefore seem to depend on increased protein translation rates, possibly to satisfy their enormous demand of resources for biomass production. Deregulation in both metabolism and protein translation are known to interfere with cell proliferation and survival. Cell cycle progression has been shown to be dependent on the metabolic and nutritional status of a cell[57,58] and impaired mitochondrial bioenergetics alone are capable of inducing $G_1$/S cell cycle arrest in *Drosophila*[59]. Furthermore, ribosome biogenesis stress and increased levels of the ribosomal protein L5 (Rpl5) could contribute to the accumulation of p53 protein in *Yy1cKO* cortices, similar to observations made in other systems[60,61]. Thus, the concurrent attenuation of several metabolic pathways as well as decreased protein translation is likely too harsh to cope with for fast proliferating NPCs and results in the upregulation of stress sensors, proliferation defects and p53-dependent cell death observed in *Yy1cKO* embryos.

Despite its constitutive expression in NPCs and neurons, our results show that Yy1 becomes less important over the course of embryonic neurogenesis. Our data indicate that, overall, Yy1 binds to and regulates similar target genes throughout cortex development. In agreement, reconstruction of the Yy1 gene regulatory network revealed that E2F and ETS family members were the co-factors with the highest enrichment score at both developmental stages. However, a small subset of genes were Yy1-dependent at early stages but not altered upon late ablation of Yy1. Some genes of this group have already been implicated in processes contributing to the Yy1-dependent phenotype, including oxidative phosphorylation (Ndufc1[62], Ndufb4[62], Ndufb5[62]),

mitochondrial biogenesis (Ppargc1a[63,64]), ribosome subunit and activator of p53 (Rpl5[61]) and inhibition of protein translation (Akt1s1[65]). Judged by their function, other genes could potentially also influence cell behavior by modulating lipid metabolism (Napepld) and protein translation (Eif1b, Rps13). Therefore, it is conceivable that rescued expression of these genes in late-stage *Yy1iKO* cortices may contribute to the ameliorated metabolic phenotype observed. Surprisingly, many cell cycle progression genes, although not directly bound by Yy1, were upregulated specifically upon late ablation of Yy1. This further emphasizes stage-specific transcriptional responses to Yy1 inactivation that potentially influence the susceptibility of NPCs toward loss of Yy1 at different developmental stages.

In the complex and highly dynamic environment of brain development, several cellular processes affect corticogenesis in a stage-dependent manner. Cell cycle length of NPCs generally decreases as the embryo develops[66–68]. As a result of their fast proliferative nature and their substantial biosynthetic demand, early NPCs are thought to be especially vulnerable to cellular stress. For example, genome integrity is vital for NPCs to give rise to healthy daughter cells, but the susceptibility of NPCs to genomic stress appears to alter during the course of development. Interference with genome integrity by ablation of the DNA replication factor TopBP1 resulted in more pronounced neurodevelopmental defects when induced in early, compared with late NPCs, despite causing comparable degrees of DNA damage at both developmental stages[69,70]. Furthermore, mRNA expression profiling of RNA-binding proteins identified many genes to be differentially regulated with progressive cortex development. Consistent with the high protein-synthesis rate of early NPCs, the expression of a majority of genes involved in RNA binding, protein translation, and ribosomal proteins is higher at early stages of corticogenesis[71]. Conceivably, this reflects the high biosynthetic demand of fast proliferating, early progenitor cells in order to synthesize proteins and precursor molecules for biomass production. In analogy to the temporal difference in vulnerability to genotoxic stress, our study indicates that early NPCs are particularly susceptible to metabolic and protein synthesis stress. The shorter cell cycle might render early NPCs less flexible and increase their dependency on rapid accumulation of building blocks. Our findings thus show that temporal control of basic cellular processes is critical for brain development. In sum, Yy1 plays a central role in cortex development by influencing stage-dependent progenitor cell behavior through transcriptional regulation of metabolism and protein synthesis.

## Methods

**Animal models**. All animal experiments were conducted in accordance with Swiss guidelines and approved by the Veterinary Office of the Canton of Zurich, Switzerland. Previously described *Yy1*[lx/lx] mice[17] were crossed to *Emx1-Cre*[16] or *Emx1-CreER*[T2] mice[22] to ablate *Yy1* function in the developing cortex (*Emx1-Cre Yy1*[lx/lx], *Emx1-CreER*[T2] *Yy1*[lx/lx]). The following genotypes were used as control animals: *Emx1-Cre Yy1*[lx/wt], *Emx1-Cre Yy1*[wt/wt], *Yy1*[lx/lx], and *Emx1-CreER*[T2] *Yy1*[lx/wt], *Emx1-Cre ER*[T2] *Yy1*[wt/wt]. To study the function of p53 in the context of *Yy1* ablation, previously described *Trp53*[lx/lx] mice[27] were crossed to *Emx1-Cre Yy1*[lx/lx] mice (*Emx1-Cre Yy1*[lx/lx] *Trp53*[lx/lx]). *Emx1-Cre Yy1*[lx/wt] *Trp53*[lx/lx] served as control animals. All genotypes were present at Mendelian ratios and control animals showed no overt phenotype. Wild-type NMRI or Swiss mice (Janvier, France) were used for ChIP-seq, knockdown experiments, to determine mRNA levels and protein translation rates at different developmental stages. To generate embryos of a certain developmental stage, mice were mated overnight and the next morning was defined as E0.5. Timed ablation of *Yy1* in *Emx1-CreER*[T2] *Yy1*[lx/lx] mice was induced by intraperitoneal (i.p.) injection of TM (200 mg/kg body weight, Sigma) into pregnant females at indicated stages. Pharmacological inhibition of p53 was achieved by i.p. injection of pregnant females with 2.2 mg/kg body weight PFTα[29] (Enzo Life Sciences) twice per day from E10.5–E12.5.

**Cell culture and siRNA transfection**. NPCs from dorsal E11.5 cortex (NMRI mice) were isolated by microdissection of the dorsal cortex in Hank's Balanced Salt Solution (HBSS) (14170, Gibco) and digestion in HBSS (14170, Gibco) containing 0.35 mg/ml collagenase type 3 (M3D14157, Worthington) and 0.04% Trypsin (7001719, Life Technologies). The digestion mix was inactivated using soybean trypsin inhibitor (Sigma). Cells were grown on plates (Corning) coated with poly-L-lysine (Cultrex) and fibronectin (Sigma) in Dulbecco's Modified Eagle Medium (DMEM)/F12+ Glutamax medium, supplemented with B-27, N2 (all from Gibco), 20 ng/ml basic fibroblast growth factor (bFGF), and 20 ng/ml EGF (both Peprotech) (cortex medium). For RNA interference of Yy1, cells were transfected with control siRNA (Medium GC Duplex #2, Invitrogen) or siRNAs against Yy1 (Stealth RNAi, Invitrogen) (Supplementary Table 1) using JetPRIME transfection reagent (Polyplus). After 48 h of siRNA treatment, cells were harvested using Trypsin (Gibco) and soybean trypsin inhibitor (Sigma), washed with phosphate-buffered saline (PBS) (Gibco) and processed for further experiments.

**Immunohistochemistry**. Embryo heads or E18.5 brains were dissected, washed in HBSS (Gibco) and fixed overnight in 4 % paraformaldehyde at 4 °C, followed by dehydration in ethanol and paraffin embedding. Sagittal 5-μm paraffin sections were deparaffinized, high-pressure antigen retrieval in citrate buffer (pH 6) was performed, and sections were subsequently stained following standard protocols (washes: PBS or PBS-TritonX 0.2% (PBS-T); blocking solution: 1% bovine serum albumin (BSA) in PBS-T). The following primary antibodies were used (diluted in blocking solution): anti-cleaved caspase 3 (rabbit Cell signaling 9661, 1:300), anti-Ctip2 (rat Abcam ab18465, 1:200), anti-CyclinB1 (rabbit Santa Cruz sc-752, 1:200), anti-CyclinD1 (mouse Santa Cruz sc-450, 1:50), anti-Dcx (guinea pig Millipore ab2253, 1:300), anti-p53 (rabbit Santa Cruz sc-6243, 1:50; mouse Cell Signaling, 1:300), anti-Pax6 (mouse DSHB, 1:50 and rabbit Covance PRB-278P, 1:200), anti-phospho Histone 3 (mouse PH3 Cell Signaling 9706, 1:300), anti-Reelin (mouse Novus Biological NB600-1081, 1:100), anti-Sox2 (rabbit Chemicon AB5603, 1:100 and mouse R&D MAB2018), anti-Tbr1 (rabbit Abcam ab31940, 1:200), anti-Tbr2 (rabbit Chemicon AB9618, 1:200), anti-Yy1 (rabbit sc1703 and mouse sc-7341, both Santa Cruz, 1:100).

Secondary isotype-specific Alexa488-, Alexa546-, and Alexa647-conjugated antibodies against mouse were from Invitrogen, secondary biotin anti-rabbit, steptavidin-, Dylight488-, and DyLight549- conjugated antibodies against other species were purchased from Jackson ImmunoResearch. Secondary antibodies were diluted 1:500 in blocking solution and incubated for 1 h at RT. Nuclei were counterstained with 4′,6-diamidino-2-phenylindole (DAPI, Sigma, 1:2000).

**Immunoblotting**. Cells or cortex tissue were harvested and lysed in radio-immunoprecipitation buffer (RIPA) buffer (89900, Thermo Scientific) containing Halt Phosphatase and Protease Inhibitor Cocktail (78420, 87786, Thermo Scientific). Cortex tissues were homogenized and both, cells, and tissue samples were sonicated in RIPA buffer. Sodium dodecyl sulfate polyacrylamide gel electrophoresis (SDS-PAGE) was carried out on 4–20% Mini-PROTEAN TGX Gels (456–1094, Bio Rad). The following primary antibodies (4 °C, O/N, in Odyssey blocking buffer (927–40000, LI-COR Biosciences)) were used: anti-β-actin (mouse Sigma a5316, 1:10000), anti-CyclinB1 (rabbit Santa Cruz sc-752, 1:500), anti-CyclinD1 (mouse Santa Cruz sc-450, 1:500), anti-Histone H3 (mouse Cell Signaling, 1:1000), anti-Yy1 (rabbit Santa Cruz sc1703, 1:500). Blots were stained with isotype-specific secondary antibodies in Odyssey blocking buffer for 1 h at room temperature. Blots were scanned and quantified with an Odyssey imaging system (LI-COR Biosciences).

**Flow cytometry**. For assessment of cell death, cells were harvested, washed, and stained with Annexin V-Cy5 according to the manufacturer's instructions (BD Bioscience). For cell cycle analysis, cells were pulsed with EdU (10 μM) for 1 h, harvested and fixed with 4 % paraformaldehyde. Staining for EdU incorporation was carried out as indicated by the manufacturer's instructions (Click-It EdU Flow

Cytometry kit, Thermo Scientific) and the DNA content was visualized with propidium iodide. For all flow cytometry experiments, 10,000–50,000 cells were recorded. All samples were analyzed on a BD FACS Canto II flow cytometer. FlowJo v7.6.5. was used to process the obtained data.

**Quantitative real-time PCR and RNA-sequencing**. Dorsal cortex tissue was dissected and total RNA was isolated with the RNAeasy kit (Qiagen) and RNase-Free DNase Set (79254, Qiagen) following the manufacturer's instructions. For quantitative real-time PCR, RNA was reverse transcribed with the Maxima First Strand cDNA Synthesis kit (Fermentas, K1641). Quantitative real-time PCR was carried out using LightCycler® SYBR Green I master mix (Roche, 4887352001) and was run on a LightCycler® 480 System (Roche). Each experiment was performed in a minimum of biological and technical triplicates. Obtained data were analyzed by the ΔCt method and normalized to the expression levels of β-actin for comparing control and *Yy1cKO* samples or normalized to the average expression of four housekeeping genes (β-actin, Mrps6, Tbp, Ywazh) for comparison of different embryonic stages. Primers used are listed in Supplementary Table 2.

For RNA-sequencing at E11.5, total RNA was isolated of control (n = 3, from two different litters) and *Yy1cKO* (n = 3, from two different litters) dorsal cortex tissue. Sequencing was performed at the Functional Genomics Center Zurich, Switzerland, using the Illumina HiSeq 2000 platform. The heatmap in Fig. 4a shows differentially expressed genes with |log2 fold change| > 0.3, p < 0.05, FDR < 0.01. GO networks were constructed with ClueGO (Version 2.3.2) and Cytoscape (Version 3.4.0). Clusters with less than 3 nodes were omitted.

For RNA-sequencing at E15.5, cells of control (n = 3) and *Yy1iKO* (n = 3) dorsal cortex were isolated, stained for Prominin1 (rat CD133-FITC 1:100, 11-1331, eBioscience) in HBSS (Gibco) supplemented with 5% FBS for 30 min at 4 °C. Cells were washed with HBSS + 5% FBS and Prominin1-positive cells were sorted directly in cold RLT lysis buffer (Qiagen) + 1% 2-mercaptoethanol. Unstained cells and isotype-specific IgG-FITC labeled cells served as negative control for gating. Isolated RNA was sequenced at the iGE3 Genomics Platform (Geneva, Switzerland) using the Illumina 4000 platform. RNA-seq data sets are deposited at the European Nucleotide Archive (ENA) (accession number PRJEB21545 and PRJEB30271).

For the determination of mitochondrial versus genomic DNA content, E12.5 dorsal cortex tissue was dissected and total DNA was isolated with the QiaAmp DNA kit (Qiagen) following the manufacturer's instructions. Relative copy numbers of mitochondrial DNA was determined by quantitative real-time PCR for mitochondrial and genomic genes (Primers: Supplementary Table 3).

**ChIP-sequencing**. ChIP material was prepared as previously described[72]. In brief, chromatin from dissected dorsal cortex tissue of E12.5 and E15.5 NMRI mice was prepared and immunoprecipitated according to the manufacturer's instructions (Magna A/G ChIP Kit, Millipore). Crosslinked protein–DNA complexes of ~ 300–700 bp size were immunoprecipitated using 5 μg of rabbit anti-Yy1 antibody (sc1703, Santa Cruz) or 5 μg of rabbit IgG antibody (Santa Cruz). After recovery of the DNA fragments, binding sites were determined by sequencing at the iGE3 Genomics Platform, Geneva, Switzerland, using the Illumina HISeq4000 platform.

**Data analysis–peak calling**. All sequenced reads were mapped using Bowtie 2 (http://bowtie-bio.sourceforge.net/bowtie2/index.shtml) onto the UCSC mm10 reference mouse genome. The command "findPeaks" from the HOMER tool package (http://homer.salk.edu/homer/) was used to identify enriched regions in the Yy1 immunoprecipitation experiments using the "-style = factor" option (routinely used for transcription factors with the aim of identifying the precise location of DNA-protein contact). Input samples were used as enrichment-normalization control. Peak calling parameters were adjusted as following: L = 2 (filtering based on local signal), F = 2 (fold change in target experiment over input control). The annotation of peaks' position (i.e., the association of individual peaks to nearby annotated genes or genomic loci) was obtained by the all-in-one program called "annotatePeaks.pl". Finally, the HOMER program "makeUCSCfile" was used to produce bedGraph formatted files that can be uploaded as custom tracks and visualized in the UCSC genome browser (http://genome.ucsc.edu). GO networks were constructed with ClueGO (Version 2.3.2) and Cytoscape (Version 3.4.0). Clusters with less than three nodes were omitted. Computational prediction of co-factors was performed by i-cisTarget[32]. ChIP-Seq data sets are deposited at the European Nucleotide Archive (ENA) (accession number PRJEB21635).

**Metabolomics**. Isolated NPCs from E11.5 cortices were transfected with siCtrl or siYy1 (n = 9, each). After 48 h, cells were harvested and washed twice with 75 mM ammonium carbonate, ph 7.4, and snap frozen in liquid nitrogen. Metabolites were extracted two times with cold acetonitrile-methanol-water (40:40:20 ratio, − 20 °C). Extracted metabolites were analyzed by flow injection–time of flight mass spectrometry on an Agilent 6550 QTOF instrument operated in the negative mode as previously described[73]. Detectable ions were putatively annotated by matching measured mass-to-charge ratios with theoretical masses of compounds listed in the human metabolome database v3.0[74] using a tolerance of 0.001 amu. Pathway definition of differentially abundant metabolites was performed with the Small Molecule Pathway Database[75]. P values were calculated by two-tailed, heteroscedastic t test and

were adjusted for FDR according to the Benjamini–Hochberg procedure. All calculations were done in Matlab (The Mathworks, Natick, MA).

**Oxygen consumption measurement**. Cells from dorsal E12.5 or E15.5 cortex were isolated and equal numbers of living cells (determined by Trypan Blue exclusion assay) were seeded into Seahorse XFᴾ or XFᵉ²⁴ cell plates (Seahorse Bioscience). Cells were incubated for 3 h at 37 °C and 5% $CO_2$ in cortex medium to allow attachment to the poly(L-lysine) (PLL)/fibronectin-coated plates. To prepare the cells for OCR measurement, cells were incubated for 1 h at 37 °C with athmospheric $CO_2$ levels in non-buffered XF Base DMEM minimal medium supplemented with 0.5 mM sodium pyruvate, 5.5 mM glucose, and 2 mM glutamine (Seahorse medium). Cell viability prior to measurement was ensured by staining for 7AAD (BD Bioscience) in an independent experiment. OCR was measured on Seahorse XFᴾ or XFᵉ²⁴ extracellular flux analyzers in Seahorse medium. Relative OCR was assessed upon the injection of 1 μM oligomycin, 1 μM FCCP and 0.5 μM antimycin A/rotenone (all from Seahorse bioscience) and normalized to protein content (bicinchoninic acid assay) and the first OCR measurement of control cells. For each condition, at least six embryos from at least two different litters were measured ($n \geq 6$).

**Measurement of protein translation rate**. Cells were directly isolated from dorsal E12.5 or E15.5 cortex and equal numbers of cells were seeded into PLL/fibronectin-coated 96-well plates. Cells were incubated for 2 h at 37 °C in cortex medium to allow attachment to the plates. Alternatively, cells isolated from E11.5 cortex were treated with siRNA for 48 h as described above. For measurement of protein synthesis rates, cells were pulsed with OPP (20 μM, Thermo Scientific and Jena Bioscience) for 30 min, harvested and fixed with 4 % paraformaldehyde. Staining for OPP incorporation was carried out as indicated by the manufacturer's instructions (Click-It OPP Flow Cytometry kit, Thermo Scientific) and the DNA content was visualized with propidium iodide to distinguish between $G_0/G_1$ (DNA content = 2c) and $S/G_2/M$ (DNA content > 2c). For analysis of protein synthesis rates of Prom1+ cells, fixed OPP-pulsed cells were stained for Prominin1 (rat CD133 1:250, 14-1331-80 eBioscience; Dylight488 anti-rat 1:500, 112-486-072, Thermo Scientific) in PBS + 1% BSA (each for 30 min at 4 °C) before OPP Click-It staining. The samples were analyzed on a BD FACS Canto II flow cytometer. FlowJo (Version 7.6.5) was used to process the obtained data. The relative rate of protein translation was calculated by using the mean fluorescent intensity relative to control samples after subtraction of background staining and non-OPP-incorporating dead cells. For each condition, cells from at least five embryos from at least two different litters were measured ($n \geq 5$).

**Imaging, quantification and statistical analysis**. Epifluorescence and confocal micrographs were taken with a Leica DMI6000 B or a CLSM Leica SP8 upright microscope, processed with Adobe Photoshop or Imaris and quantified manually using ImageJ. For each quantification, at least three mutants and three controls from at least two different litters, and at least three sections per animal at different rostro-caudal levels were analyzed. A radial unit was defined as a 100 μm-wide stripe measured at the ventricular surface with perpendicular borders given by the direction of RG cell fibers. Numbers of pHH3-positive cells were counted along the whole VZ length and normalized to 600 μm VZ length. Measurements of cortical thickness, length, and width were done with ImageJ. No statistical methods were used to pre-determine sample size. Animals were not randomized and researchers were not blinded during experiments and for quantifications. Statistical analysis of all experiments was performed in Microsoft Excel using unpaired, two-tailed Student's $t$ test, or in GraphPad Prism using analysis of variance Tukey's multiple comparison test for comparing more than two groups. Statistical significance cutoff threshold was set at $p < 0.05$. Results are shown as mean ± standard deviation, except where indicated.

**Reporting summary**. Further information on research design is available in the Nature Research Reporting Summary linked to this article.

## Data availability

The sequencing data sets generated and analyzed during the current study are available in the European Nucleotide Archive (https://www.ebi.ac.uk/ena) and are accessible through the accession numbers PRJEB21545, PRJEB30271, and PRJEB21635. All relevant data that support the findings of this study are available from the corresponding author upon reasonable request. Source data for all figures are provided as a Source Data file.

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

## Acknowledgements

We thank Dr. Wolfgang Langhans and Dr. Rosmarie Clara from the ETH Zurich for their support with the Seahorse XPe24 Extracellular Flux Analyzer. We thank past and present members of the Lukas Sommer laboratory for critical comments. S.V. is supported by the Zurich University Research Priority Program "Translational Cancer Research". This work was supported by the Swiss National Science Foundation.

## Author contributions

L.Z. designed the study, conducted the experimental work, analyzed the data, and wrote the paper. S.V. helped designing the study, conducted experimental work, analyzed the data, and edited the manuscript. S.G., A.K., J.H., R.B., C.C., and N.Z. conducted experimental work and analyzed the data. M.Z. conducted experimental work and edited the manuscript. Z.K.A. analyzed the data. K.B. and N.Z. acquired funding resources. L.S. supervised the project, acquired funding resources and wrote the paper.

## Additional information

**Competing interests:** The authors declare no competing interests.

