## [Peer Review File · Nature Communications]

Reviewers' Comments:

Reviewer #1:

Remarks to the Author:

The authors examine here the function of the transcription factor Yy1 in cerebral cortex development. They report an interesting early sensitivity to Yy1 deletion, while later ablation at around mid-neurogenesis has little effect. The main phenotypes upon early deletion are cell death and proliferation deficits. While cell death is p53-dependent and can be rescued by p53 deletion or pharmacological block, the cell cycle defects still persisted. Using transcriptome and ChIP-seq analysis the authors then demonstrate a key role of Yy1 in regulating metabolism, mitochondrial function and translation. Indeed, early Yy1-deficient neuroepithelial cells have impaired respiration rates, as well as protein translation rates. Most importantly, the authors then demonstrate that these changes do not occur at later stages when translation and respiration rates are not affected by Yy1 deletion (see however comment below). Indeed, the authors demonstrate that glycolysis and translation rate normally decrease during development, thereby explaining why deletion of Yy1 at later stages has less impact. These are very interesting and important data highly relevant for the field of stem cell research as the authors identified a key role of a master regulator of metabolism in early stem cells. Moreover, the analysis of the mutant phenotype as well as genome-wide data are very carefully done.

I have only one major set of questions that deal with cell type-specificity at later stages.

- 1) While all analysis done at E11.5 examines a virtually homogenous population of neuroepithelial stem cells, the analysis of translation and respiration rates (seahorse) at E15.5 comprise a mixed population. It is essential, however, to determine to which extent these phenotypes (translation and respiration rates) are normalized in the E15.5 stem cells or are only normalized because the neuronal population that should comprise 50% of all cells at this stage is not affected. The authors could readily assess this by isolating the stem cells using prominin-based fluorescent activated cell sorting.
- 2) The same is the case for the developmental analysis – do translation and respiration rates decrease in prominin+ stem cells, or only decrease because other populations have lower rates.
- 3) A major switch in neurogenesis occurs when intermediate progenitors increase that also ease the metabolic transition of radial glial cells with high glycolytic rates to neurons with a largely oxidative metabolism. It would therefore be very interesting to know whether these intermediate progenitor cells would react differently to Yy1 deletion as the stem cells. To address this important issue the authors should discriminate between apical and basal mitosis, both in the cortices with early Yy1 deletion where proliferation seems to normalize at later stages (is this the case only due to compensation by basal progenitor mitoses?) and in cortices with later Yy1 deletion.

Reviewer #2:

Remarks to the Author:

In this manuscript, Zurkirchen et al. reported a novel function of Yy1 on mouse cortical development in a stage-dependent manner. By regulating a wide range of metabolic pathways and protein translation, Yy1 maintains proliferation and survival of NPCs at early stages of brain development, but less influence at late stages. The manuscript is well organized and written. The data is very convincing and logical. Meanwhile, I have several concerns about this study.

1. Apical and basal progenitors are included in mice dorsal cortex. Whether the two kinds of progenitors have the same sensitivity to Yy1 is very interesting. Since at E12.5, the Sox2+ apical progenitors are predominant and at E15.5 Tbr2+ basal progenitors are abundant. Is the less-dependent at late stages caused by less sensitivity of basal progenitor to Yy1?
2. With sophisticated experiments, the authors found that Yy1 was also involved in cell cycle arrest

and cell survival by modulation of metabolic process and protein translation. In OP-puro assay, the protein synthesis rate was highly reduced in Yy1 mutant. As Yy1 is overexpressed in some cancers, I believe it will be very interesting to check whether Yy1 is sufficient to accelerate protein synthesis and cell cycle in neural progenitors which will make the conclusion much stronger.

3. In Figure 1 experiment, the author used different staining, that is the caspase3, CyclinD1 and CyclinB1 to indicated Yy1 improving cell survival and proliferation, however, the co-staining of caspase3 and CyclinD1 would be more directly illustrate the decreasing population of NPCs.

4. In Figure 2 experiment, the author injected tamoxifen in Emx-CreERT2Yy1lx/wt to define the deletion stage of Yy1 during the development. The results showed that ablation of Yy1 at E12.5 affected proliferation and cell death less prominently at E14.5, comparing the distinguishable contrast in E12.5 after injection at E9.5. It seemed that the Yy1-ablation induced cellular phenotype would be mild around E14.5 regardless of the early deletion (E9.5) or the late deletion (E12.5). The author would be better also check the cellular phenotype at E14.5 after deletion at E9.5. Additionally, how is the situation of loss-of-function of Yy1 between E9.5 and E12.5 (E10.5? E11.5?), to define the critical window of Yy1 to regulate the microcephaly phenotype?

5. In Figure 3, the Yy1Trp53dKO mice could partially rescue the Yy1-p53-induced defect on proliferation. Apart from p53, Is there any other interacting gene responsible for the Yy1-induced microcephaly phenotype, according to the sequencing data?

Minor comments:

1. In Figure1k and Figure 2p, based on current images, the difference is dramatic, pls replace a representative image if the authors claimed the different is not significant.
2. A schematic diagram will help the readers to understand the paper better.

Reviewer #3:

Remarks to the Author:

Zurkirchen and colleagues investigate the role of the transcription factor Yy1 in mouse embryonic cortex development. Deletion of Yy1 from early embryonic stages (E10.5), induces cell cycle arrest and apoptosis in cortical cells at the onset of cortical neurogenesis (E12.5), but not at later stages (E15.5). Moreover stage-specific induction of Yy1 inactivation reveals that Yy1 is indispensable for cortical cell survival at early not at middle stages of cortical neurogenesis. Then, RNAseq analysis of mutant mice at early neurogenesis stage indicate alterations of metabolic pathways and mitochondria-related genes as well as protein translation-related genes at early neurogenesis stage (E12.5), which are confirmed functionally using metabolic and translation analysis assays. These effects appear to be directly transcriptional as chipseq analyses at the same stage reveals a large overlap of potential direct target genes with those downregulated in the mutant mice. Finally, the authors perform chipseq at later stages, which reveal a similar pattern of binding of Yy1 than at early stages, while examination of a smaller subset of genes show downregulation but to a lesser extent than at earlier stages. Overall these data lead the authors to conclude that cortical neurogenesis is sensitive to the loss Yy1 only at early stages reflecting different demands on metabolic profile or protein translation. Overall this is a very thorough and creative study on an original and timely topic, ie the impact of metabolic control on neural development. However there remain some issues to address for the study to be fully convincing and meaningful.

- The authors reveal similarity and differences of phenotypes of Yy1 inactivation at the two developmental stages. However, it is not clear how these stage-specific Yy1 effects are achieved. As transcriptional downregulation appears to be less affected at late than early stages could it be that another co-transactivating factor is differentially expressed in early and not later stages? Are there any indications from the chipseq data of enrichment of sequence data for other trans factors next to

Yy1-dependent genes showing discrepancy between Yy1 binding and expression changes?

- Deletion of both Yy1 and Trp53 can rescue the phenotype of Yy1-dependent apoptosis of cortical cell but not cell cycle arrest, and the double KO mice show reduced cortical size (Fig. 3d,e). These results suggest that Yy1-dependent control of cell cycle or proliferation has a key impact on corticogenesis: but what is it? Again, deeper analysis of the rnaseq and chipseq data could reveal candidate genes linking Yy1 and cell cycle arrest and thereby provide more mechanistic insights on the causality of the microcephaly observed.

- In Fig. 6, the authors measured OCR using Yy1cKO cells. Cell viability is essential for interpreting the result of OCR assays. The authors mention about living cell preparation method, but they do not mention about viability just before OCR assay. They should compare cell viability between control and Yy1cKO cell under these experimental conditions, to make sure that the differences observed genuinely reflect metabolic changes and not health/viability.

- The authors interpret their findings by differential sensitivity of early vs late progenitors to Yy1 loss and metabolic/translation activity. One major difference between these stages is the relative abundance of intermediate/basal progenitors: the authors should check whether these progenitors are somehow more resistant by determining their proportion at early and late stages, for instance using Tbr2 immunostainings.

Reviewer #4:

Remarks to the Author:

In this manuscript, Zurkirchen and colleagues analyzed mice conditionally mutant for the Yin Yang 1 (YY1) gene in the developing cerebral cortex and further performed RNA-Seq and ChIP-Seq studies to examine the molecular mechanism of its action. They report that YY1 regulates cerebral cortex development by acting in a stage-specific manner to regulate the proliferation and survival of neural progenitor cells (NPCs) and this requirement is due to a role in regulating a wide range of metabolic genes and protein synthesis. While the findings are potentially of interest to Nature Communications readers, the key conclusions are not fully supported by the data presented as they could easily be due to technical flaws. The manuscript is, therefore, not of sufficient vigor and quality for publication in this journal.

Major Issues

(1) One key conclusion is that YY1 plays a stage-specific role in regulating brain development by showing that ablation of YY1 at different stages (e.g. E9.5 vs. E12.5) had different effect on NPCs. However, there are equally plausible, alternative explanations for the data presented.

(a) Emx-IRES-Cre is not specific to NPCs, and, therefore, it is quite possible the effect on NPCs after ablation around E9.5 (cKO) is not due to a requirement for YY1 in NPCs during early neurogenesis but is secondary to defects elsewhere. Ablation of YY1 at E12.5 using CreERT2 (iKO) – showing a much milder effect – might simply have allowed the authors to bypass the effect on non-NPCs at earlier developmental stages. In other words, the data presented could be interpreted as a non-essential role for YY1 in NPC proliferation and survival.

(b) It takes time for tamoxifen to induce gene deletion and for YY1 proteins produced before that to be fully depleted. So it is not surprising for NPC phenotypes to be weaker.

(c) Since the focus of this study is on a role for YY1 in NPCs, it is puzzling why the authors did not use hGFAP-Cre, which is available and highly specific for cortical neural stem cells, to ascertain such a role.

(2) Another key conclusion by the authors is that YY1 plays developmental stage-specific roles in brain development by regulating genes in a developmental stage-specific manner. They performed RNA-Seq and ChIP-Seq experiments using tissues from E11.5/E12.5 and E15.5 and discovered changes in expression of genes potentially regulated by YY1, including those in the metabolic pathway. The data reported could also be interpreted by the alternative possibilities discussed above.

(a) Changes in gene expression between E11.5 and E15.5 may simply reflect the effectiveness of cKO and iKO in generating the mutant cells for analysis and the non-cell-autonomous effect on NPCs. No evidence is shown whether the changes are a cause for, or a consequence of, changes in NPC behavior.

(b) The above interpretation is further supported by the observation from ChIP-Seq experiments that YY1-binding to its target genes showed only minor differences between E12.5 and E15.5.

Minor Points:

(1) Rescue experiments should be performed in assays using cultured cells, including OCR and OP-puro (OPP) intensity measurements.

(2) The RNA-seq and ChIP-Seq data should be better analyzed so that readers could have a better appreciation of the consistency of the two datasets and be able to distinguish direct and indirect targets of YY1.

(3) The phrase – that "A key feature of cortex development is the gradual decrease in the cell cycle length of NPCs as the embryo develops" – on page 16 is incorrect.

Reviewers' comments:

Reviewer #1 (Remarks to the Author):

The authors examine here the function of the transcription factor Yy1 in cerebral cortex development. They report an interesting early sensitivity to Yy1 deletion, while later ablation at around mid-neurogenesis has little effect. The main phenotypes upon early deletion are cell death and proliferation deficits. While cell death is p53-dependent and can be rescued by p53 deletion or pharmacological block, the cell cycle defects still persisted. Using transcriptome and ChIP-seq analysis the authors then demonstrate a key role of Yy1 in regulating metabolism, mitochondrial function and translation. Indeed, early Yy1-deficient neuroepithelial cells have impaired respiration rates, as well as protein translation rates. Most importantly, the authors then demonstrate that these changes do not occur at later stages when translation and respiration rates are not affected by Yy1 deletion (see however comment below). Indeed, the authors demonstrate that glycolysis and translation rate normally decrease during development, thereby explaining why deletion of Yy1 at later stages has less impact. These are very interesting and important data highly relevant for the field of stem cell research as the authors identified a key role of a master regulator of metabolism in early stem cells. Moreover, the analysis of the mutant phenotype as well as genome-wide data are very carefully done.

We greatly appreciate reviewer #1's positive and encouraging comments and have addressed the raised points in the revised version of the manuscript to improve the study.

I have only one major set of questions that deal with cell type-specificity at later stages.

1) While all analysis done at E11.5 examines a virtually homogenous population of neuroepithelial stem cells, the analysis of translation and respiration rates (seahorse) at E15.5 comprise a mixed population. It is essential, however, to determine to which extent these phenotypes (translation and respiration rates) are normalized in the E15.5 stem cells or are only normalized because the neuronal population that should comprise 50% of all cells at this stage is not affected. The authors could readily assess this by isolating the stem cells using prominin-based fluorescent activated cell sorting. 2) The same is the case for the developmental analysis – do translation and respiration rates decrease in prominin+ stem cells, or only decrease because other populations have lower rates.

Response to comments 1 and 2: We fully agree with the important issues brought up by the reviewer. Therefore, as suggested by the reviewer, we have now used indirect flow cytometry in combination with OP-puro labelling to assess protein synthesis rates in E15.5 control and *Yy1**lKO* Prominin1-positive radial glia cells (Fig. 8h,j). Unlike at E12.5 (Fig. 6j-m), late ablation of *Yy1* did not result in decreased protein synthesis rates in E15.5 Prominin1-positive cells, indicating a stage-dependent susceptibility of NPCs to *Yy1* loss.

Furthermore, as suggested by the reviewer, we have used a similar approach to demonstrate that protein translation rates indeed decrease during development in Prominin1-positive cells (Fig. 8o,p). In addition, we have isolated Prominin1-positive cells by FACS and performed RNA-Sequencing comparing control and *Yy1**lKO* radial glia cells at E15.5 (Fig. 7g). These new data reveal that *Yy1* controls expression of a very similar set of genes at E11.5 and E15.5, although some stage-specific *Yy1*-dependent gene expression was observed (see new Fig. 7h and Suppl. Fig. 10). The latter could potentially also contribute to the observed phenotypic differences at E11.5 vs. E15.5, as now discussed on page 18.

We attempted to measure OCR using the Seahorse Extracellular flux analyzer in FACS-isolated E15.5 Prominin1-positive live cells. Unfortunately, the sorting procedure turned out to be too harsh and resulted in max. 50% alive cells, precluding exact determination of OCR directly after isolation. As mentioned by reviewer #3, OCR measurement is sensitive to cell death. Therefore, the low amount of viable cells and their likely contamination with dead/dying cells did not allow us to measure OCR in Prominin1-positive cells.

3) A major switch in neurogenesis occurs when intermediate progenitors increase that also ease the metabolic transition of radial glial cells with high glycolytic rates to neurons with a largely oxidative metabolism. It would therefore be very interesting to know whether these intermediate progenitor cells would react differently to *Yy1* deletion as the stem cells. To address this important issue the authors should discriminate between apical and basal mitosis, both in the cortices with early *Yy1* deletion where proliferation seems to normalize at later stages (is this the case only due to compensation by basal progenitor mitoses?) and in cortices with later *Yy1* deletion.

Response to comment 3: We have addressed this important point, as suggested, by discriminating apical and basal pHH3+ cells and determining the ratio of apical/basal mitosis at all stages analyzed (Fig. 1f, Fig. 2h,p, Fig. 3j, Suppl. Fig. 1k and Suppl. Fig. 3d,h). The newly implemented data show that *Yy1* is not selectively required in one type of progenitor cell, as both apical and basal pHH3+ cells were similarly affected and their percentage was not changed compared to control embryos. Thus, the lack of difference between control vs.

Yy1iKO overall mitotic cells at later stages is not due to compensation by basal progenitor mitoses (see also response to reviewer #2 comment 1).

Reviewer #2 (Remarks to the Author):

In this manuscript, Zurkirchen et al. reported a novel function of Yy1 on mouse cortical development in a stage-dependent manner. By regulating a wide range of metabolic pathways and protein translation, Yy1 maintains proliferation and survival of NPCs at early stages of brain development, but less influence at late stages. The manuscript is well organized and written. The data is very convincing and logical.

Response: We would like to thank reviewer #2 for the very positive comments. We have carefully addressed the points mentioned and feel that the revised version of the manuscript is significantly strengthened.

Meanwhile, I have several concerns about this study.

1. Apical and basal progenitors are included in mice dorsal cortex. Whether the two kinds of progenitors have the same sensitivity to Yy1 is very interesting. Since at E12.5, the Sox2+ apical progenitors are predominant and at E15.5 Tbr2+ basal progenitors are abundant. Is the less-dependent at late stages caused by less sensitivity of basal progenitor to Yy1?

Response: We acknowledge the importance of discriminating different progenitor subpopulations and have determined the number of Pax6+ (apical progenitors), Tbr2+ (intermediate progenitors) and cells transitioning from apical to intermediate progenitors (Pax6+/Tbr2+) for each of the stages and genotypes analyzed (Suppl. Fig. 4). The new data show that while the absolute number of Pax6+ and Tbr2+ cells decrease upon knockout of Yy1 at early stages, the percentage of different subpopulations does not change. Therefore, we conclude that the decreased susceptibility to loss of Yy1 at later cortical stages is not caused by differential sensitivity of apical vs basal progenitor cells (see also response to reviewer #1 comment 3).

2. With sophisticated experiments, the authors found that Yy1 was also involved in cell cycle arrest and cell survival by modulation of metabolic process and protein translation. In OP-puro assay, the protein synthesis rate was highly reduced in Yy1 mutant. As Yy1 is

overexpressed in some cancers, I believe it will be very interesting to check whether Yy1 is sufficient to accelerate protein synthesis and cell cycle in neural progenitors which will make the conclusion much stronger.

We would like to thank the reviewer for this interesting comment. We, too, believe that it would be intriguing to investigate the effect of Yy1 overexpression in NPCs. We tried to overexpress Yy1 using both lentiviral transduction as well as transient plasmid overexpression with different transfection protocols in directly isolated NPCs at E11.5. Unfortunately, we experienced technical problems and were not able to achieve reliable overexpression of Yy1. We therefore cannot conclude whether overexpression of Yy1 affects protein synthesis and cell proliferation.

3. In Figure 1 experiment, the author used different staining, that is the caspase3, CyclinD1 and CyclinB1 to indicated Yy1 improving cell survival and proliferation, however, the co-staining of caspase3 and CyclinD1 would be more directly illustrate the decreasing population of NPCs.

Response: We have co-stained CyclinD1 and cCasp3 at E12.5 in control and Yy1cKO, as suggested, and have attached the images to this response letter for your perusal; however, we think that these data do not provide essential additional information and would, therefore, suggest not to include them in the main manuscript.

4. In Figure 2 experiment, the author injected tamoxifen in *Emx-CreERT2Yy1lx/wt* to define the deletion stage of Yy1 during the development. The results showed that ablation of Yy1 at E12.5 affected proliferation and cell death less prominently at E14.5, comparing the distinguishable contrast in E12.5 after injection at E9.5. It seemed that the Yy1-ablation induced cellular phenotype would be mild around E14.5 regardless of the early deletion (E9.5) or the late deletion (E12.5). The author would be better also check the cellular phenotype at E14.5 after deletion at E9.5.

Response: In the revised version of the manuscript, we have now performed Tamoxifen-mediated ablation in *Emx1-CreER^{T2} Yy1^{lx/lx}* embryos at E9.5 (new Suppl. Fig. 3), which resulted in a phenotype at E14.5 comparable to that seen in *Yy1cKO* embryos (see Fig. 1). Thus, proliferation and apoptosis in Yy1 knockout brains peak at E12.5 and then gradually decrease until E15.5 (see Fig. 1, Fig. 2, Suppl. Fig. 1, Suppl. Fig. 3). Moreover, since the susceptibility to loss of Yy1 decreases from E9.5 to E10.5 and further (see Fig. 2b,c), the defects in proliferation and apoptosis are less pronounced at E14.5 compared to E12.5.

Additionally, how is the situation of loss-of-function of Yy1 between E9.5 and E12.5 (E10.5? E11.5?), to define the critical window of Yy1 to regulate the microcephaly phenotype?

Response: We appreciate the reviewer's helpful suggestion to increase the resolution of our developmental analysis. We have implemented two additional stages and ablated Yy1 at E10.5 and E11.5 (Fig. 2a,b,c). As is evident from these new data, the effect of Yy1 ablation on cortical size is gradually decreasing with later ablation time points.

Furthermore, we have analyzed proliferation (E11.5) and apoptosis (E10.5, E11.5) in Yy1cKO embryos in Suppl. Fig. 1h-n. The first signs of the Yy1 ablation phenotype start to appear at E11.5, although it is fully established by E12.5 only.

5. In Figure 3, the Yy1Trp53dKO mice could partially rescue the Yy1-p53-induced defect on proliferation. Apart from p53, Is there any other interacting gene responsible for the Yy1-induced microcephaly phenotype, according to the sequencing data?

Response: To address this point, we compared Yy1-dependent gene expression during cortex development by means of computational prediction of co-factors with binding motifs at Yy1-bound genes. Both at E12.5 and E15.5, the highest enriched co-factor motifs are E2f and Ets transcription factor family members (Supplementary Fig. 11a,b), suggesting that the gene regulatory network dependent on Yy1 does not change along development. Therefore, as now pointed out on p.12-14 (Results section) and p. 18-19 (Discussion), these findings support our model that the stage-dependent phenotype in NPC survival and proliferation seen upon Yy1 inactivation reflects stage-dependent requirements for metabolic/biosynthetic processes.

Minor comments:

1. In Figure 1k and Figure 2p, based on current images, the difference is dramatic, pls replace a representative image if the authors claimed the different is not significant.
2. A schematic diagram will help the readers to understand the paper better.

Response: We thank the reviewer for the advice and changed the images in Fig. 1l and Fig. 2q to more representative pictures. In addition, we have added a schematic diagram summarizing the findings in Fig. 8q.

Reviewer #3 (Remarks to the Author):

Zurkirchen and colleagues investigate the role of the transcription factor Yy1 in mouse embryonic cortex development. Deletion of Yy1 from early embryonic stages (E10.5), induces cell cycle arrest and apoptosis in cortical cells at the onset of cortical neurogenesis (E12.5), but not at later stages (E15.5). Moreover stage-specific induction of Yy1 inactivation reveals that Yy1 is indispensable for cortical cell survival at early not at middle stages of cortical neurogenesis. Then, RNAseq analysis of mutant mice at early neurogenesis stage indicate alterations of metabolic pathways and mitochondria-related genes as well as protein translation-related genes at early neurogenesis stage (E12.5), which are confirmed functionally using metabolic and translation analysis assays. These effects appear to be directly transcriptional as chipseq analyses at the same stage reveals a large overlap of potential direct target genes with those downregulated in the mutant mice. Finally, the authors perform chipseq at later stages, which reveal a similar pattern of binding of Yy1 than at early stages, while examination of a smaller subset of genes show downregulation but to a lesser extent than at earlier stages. Overall these data lead the authors to conclude that cortical neurogenesis is sensitive to the loss Yy1 only at early stages reflecting different demands on metabolic profile or protein translation.

Overall this is a very thorough and creative study on an original and timely topic, ie the impact of metabolic control on neural development. However there remain some issues to address for the study to be fully convincing and meaningful.

Response: We greatly appreciate reviewer #3's positive feedback and have addressed the suggestions in the revised version of the manuscript.

- The authors reveal similarity and differences of phenotypes of Yy1 inactivation at the two developmental stages. However, it is not clear how these stage-specific Yy1 effects are achieved. As transcriptional downregulation appears to be less affected at late than early stages could it be that another co-transactivating factor is differentially expressed in early and not later stages? Are there any indications from the chipseq data of enrichment of sequence data for other trans factors next to Yy1-dependent genes showing discrepancy between Yy1 binding and expression changes?

Response: We are thankful for the constructive suggestions. To investigate co-factors of Yy1, we set up a collaboration with Stein Aerts's laboratory at the KULeuven, BE. The expertise of his group helped us identifying E2f and Ets family members as co-factors of Yy1 (Suppl. Fig. 11). Both at E12.5 and E15.5, the same co-factors were enriched in our CHIP-Seq datasets, further supporting our conclusion that Yy1 regulates a similar program at early

and later developmental stages and that the stage-dependent phenotype seen upon *Yy1* inactivation reflects stage-dependent requirements for metabolic/biosynthetic processes.

- Deletion of both *Yy1* and *Trp53* can rescue the phenotype of *Yy1*-dependent apoptosis of cortical cell but not cell cycle arrest, and the double KO mice show reduced cortical size (Fig. 3d,e). These results suggest that *Yy1*-dependent control of cell cycle or proliferation has a key impact on corticogenesis: but what is it? Again, deeper analysis of the rnaseq and chipseq data could reveal candidate genes linking *Yy1* and cell cycle arrest and thereby provide more mechanistic insights on the causality of the microcephaly observed.

Response: We have addressed this point by investigating whether *Yy1* regulates and binds to genes that are important for cell cycle progression. To this end, we evaluated the expression of cyclins, cyclin-dependent kinases (Cdk) and cell-cycle division (Cdc) genes annotated in the GO term 'regulation of cell cycle' upon knockout of *Yy1*. Suppl. Fig. 10g shows that the expression of some cell cycle genes is altered upon early knockout of *Yy1*. Importantly, however, only two genes are directly bound by *Yy1* in our ChIP-Seq datasets and they are regulated in the same direction at both developmental stages. Thus, those two genes are unlikely the cause for the rescue of the *Yy1* phenotype at later stages. Further, since the vast majority of cell cycle genes is not bound by *Yy1*, it appears that their transcriptional alterations upon loss of *Yy1* are due to indirect effects. Most conceivably, the indirect effectors are directly bound genes involved in metabolism and protein translation, as shown in Fig. 5.

- In Fig. 6, the authors measured OCR using *Yy1*cKO cells. Cell viability is essential for interpreting the result of OCR assays. The authors mention about living cell preparation method, but they do not mention about viability just before OCR assay. They should compare cell viability between control and *Yy1*cKO cell under these experimental conditions, to make sure that the differences observed genuinely reflect metabolic changes and not health/viability.

Response: We have addressed this point by assessing cell viability after incubation in Seahorse medium (1h, 37°C, atmospheric CO₂) just before OCR measurement. Both control and *Yy1*cKO wells contained on average more than 97% viable cells (7AAD-negative) (Suppl. Fig. 9b,c). We therefore conclude that the difference in OCR is indeed due to metabolic changes inflicted by loss of *Yy1*.

- The authors interpret their findings by differential sensitivity of early vs late progenitors to

Yy1 loss and metabolic/translation activity. One major difference between these stages is the relative abundance of intermediate/basal progenitors: the authors should check whether these progenitors are somehow more resistant by determining their proportion at early and late stages, for instance using Tbr2 immunostainings.

Response: We fully agree that the discrimination between different progenitor subtypes is important and have added immunostainings and quantifications for Pax6 and Tbr2 for all stages analyzed. In brief, we determined the number of Pax6+ (apical progenitors), Tbr2+ (intermediate progenitors) and cells transitioning from apical to intermediate progenitors (Pax6+/Tbr2+) for each of the stages and genotypes analyzed (Suppl. Fig. 4). The new data show that while the absolute number of Pax6+ and Tbr2+ cells decrease upon knockout of Yy1 at early stages, the percentage of different subpopulations does not change. Therefore, the decreased susceptibility to loss of Yy1 at later cortical stages is not caused by differential sensitivity of apical vs basal progenitor cells (please see also responses to comment 1 of reviewer #2 and comment 3 of reviewer #1).

Reviewer #4 (Remarks to the Author):

In this manuscript, Zurkirchen and colleagues analyzed mice conditionally mutant for the Yin Yang 1 (YY1) gene in the developing cerebral cortex and further performed RNA-Seq and ChIP-Seq studies to examine the molecular mechanism of its action. They report that YY1 regulates cerebral cortex development by acting in a stage-specific manner to regulate the proliferation and survival of neural progenitor cells (NPCs) and this requirement is due to a role in regulating a wide range of metabolic genes and protein synthesis. While the findings are potentially of interest to Nature Communications readers, the key conclusions are not fully supported by the data presented as they could easily be due to technical flaws. The manuscript is, therefore, not of sufficient vigor and quality for publication in this journal.

Response: We thank reviewer #4 for his feedback and discussed the points.

Major Issues

(1) One key conclusion is that YY1 plays a stage-specific role in regulating brain development by showing that ablation of YY1 at different stages (e.g. E9.5 vs. E12.5) had different effect on NPCs. However, there are equally plausible, alternative explanations for the data presented.

(a) *Emx*-IRES-Cre is not specific to NPCs, and, therefore, it is quite possible the effect on NPCs after ablation around E9.5 (cKO) is not due to a requirement for YY1 in NPCs during early neurogenesis but is secondary to defects elsewhere. Ablation of YY1 at E12.5 using CreERT2 (iKO) – showing a much milder effect – might simply have allowed the authors to bypass the effect on non-NPCs at earlier developmental stages. In other words, the data presented could be interpreted as a non-essential role for YY1 in NPC proliferation and survival.

Response: *Emx1Cre* is a very well established Cre driver line enabling efficient recombination to study the developing cortex. Gorski et al. (2002), which generated the *Emx1Cre* line, describe that recombination in the brain is specifically located to the dorsal cortex and is apparent from 10.5 onwards, as also observed in our own hands. Additional recombination can be seen in *Emx1*-expressing tissues outside the brain such as cranial ganglia and limb ectoderm. *Yy1cKO* embryos did not display anatomical differences visible in tissues other than the brain. Notably, the mesenchyme surrounding the forebrain and known to influence its formation at very early developmental stages was neither recombined by *Emx1Cre* nor altered in our *Yy1cKO* and *Yy1iKO* animals, as now mentioned on p.5. In addition, knockdown of *Yy1* in isolated primary NPCs recapitulated defects in proliferation and survival, also supporting a cell-autonomous, NPC-specific role for *Yy1* (Fig. 1q-t, Suppl Fig. 1o,p).

(b) It takes time for tamoxifen to induce gene deletion and for YY1 proteins produced before that to be fully depleted. So it is not surprising for NPC phenotypes to be weaker.

Response: Supplementary Fig. 3 b,n,o,q,r show efficient removal of *Yy1* protein using the *Emx1CreER^{T2}* line. Further, ablation at E12.5 induced a minor, transient phenotype at E14.5, which was not apparent anymore at E15.5. This suggests that there was sufficient time for the cellular phenotype to develop and vanish again. We feel that we carefully addressed that tamoxifen had enough time to recombine and induce a cellular phenotype. Moreover, we have now increased the resolution of our developmental analysis by implementing additional stages of *Yy1* ablation (Fig. 2a,b,c). As is evident from these new data, the effect of *Yy1* ablation on cortical size is gradually decreasing with later ablation time points. These data speak against the possibility that the observed phenotype is due to tamoxifen taking time to induce gene deletion (and thus potentially *delaying* emergence of a phenotype).

(c) Since the focus of this study is on a role for YY1 in NPCs, it is puzzling why the authors

did not use hGFAP-Cre, which is available and highly specific for cortical neural stem cells, to ascertain such a role.

Response: The *Emx1Cre* mouse line is a very well described tool to address dorsal cortex development and has been successfully used by us and other groups to study NPC behavior from E10.5 on. The suggested hGFAP-Cre line is known to induce recombination in the cortex starting only from E13.5 on (Zhuo et al., *Genesis*, 31:85-94, 2001), a time-point where NPCs are not susceptible anymore to loss of Yy1.

(2) Another key conclusion by the authors is that YY1 plays developmental stage-specific roles in brain development by regulating genes in a developmental stage-specific manner. They performed RNA-Seq and ChIP-Seq experiments using tissues from E11.5/E12.5 and E15.5 and discovered changes in expression of genes potentially regulated by YY1, including those in the metabolic pathway. The data reported could also be interpreted by the alternative possibilities discussed above.

(a) Changes in gene expression between E11.5 and E15.5 may simply reflect the effectiveness of cKO and iKO in generating the mutant cells for analysis and the non-cell-autonomous effect on NPCs. No evidence is shown whether the changes are a cause for, or a consequence of, changes in NPC behavior.

(b) The above interpretation is further supported by the observation from ChIP-Seq experiments that YY1-binding to its target genes showed only minor differences between E12.5 and E15.5.

Response: We thank the reviewer for the remarks. As correctly mentioned, our ChIP-Seq and RNA-Seq experiments show direct binding of Yy1 and transcriptional regulation of genes involved in metabolism and protein translation. This suggests that the metabolic phenotype is indeed directly mediated by the transcriptional activity of Yy1 at early stages.

Immunohistochemistry for Yy1 protein as well as qRT-PCR for Yy1 mRNA shows that Yy1 recombination is efficient even upon late injection of tamoxifen at E12.5 and E13.5 (Suppl Fig 3n,o,q,r, Fig. 8a). In addition, our newly implemented RNA-Seq dataset comparing Prominin1+ control and *Yy1iKO* at E15.5 clearly shows downregulation of Yy1 and differential expression of genes that are dependent on Yy1 at E11.5, demonstrating that Yy1 has been ablated efficiently.

Minor Points:

(1) Rescue experiments should be performed in assays using cultured cells, including OCR and OP-puro (OPP) intensity measurements.

Response: We would like to thank the reviewer for this interesting suggestion. We tried to rescue the metabolic phenotype upon knockdown of Yy1 in cultured NPCs by supplementation of metabolites of pathways regulated by Yy1 (nucleosides, N-acetyl-cysteine, carnitine, thymidine, hypoxanthine, methyl-aspartate, dimethyl ketoglutarate). The supplements, however, were never able to consistently revert defects in cell proliferation and survival. This indicates that the simultaneous alteration of multiple different metabolic pathways as a result of Yy1 inactivation prohibited the rescue by a single or a few metabolites. The protein translation phenotype may additionally hindered rescue by metabolite supplementation.

However, we have succeeded to rescue at least partially the Yy1 phenotype *in vivo*. Genetic ablation and pharmacological inhibition of p53 rescued apoptosis but not proliferation of NPCs, resulting in partial rescue of cortical size (Fig. 3 and Suppl. Fig. 5).

(2) The RNA-seq and ChIP-Seq data should be better analyzed so that readers could have a better appreciation of the consistency of the two datasets and be able to distinguish direct and indirect targets of YY1.

Response: We thank the reviewer for this helpful comment. We recognized the need for the suggested improvements and now display direct and indirect targets of Yy1 in Fig. 7h and Suppl. Fig. 10. In addition, we performed bioinformatics analyses to determine co-factors of Yy1 at E12.5 and E15.5, as now shown in Suppl. Fig. 11.

(3) The phrase – that "A key feature of cortex development is the gradual decrease in the cell cycle length of NPCs as the embryo develops" – on page 16 is incorrect.

Response: We thank the reviewer for mentioning this and have changed the sentence to: 'The cell cycle length of apical RG cells decreases as the embryo develops', referring to Calegari et al., 2005 and Arai et al., 2011.

Reviewers' Comments:

Reviewer #1:

Remarks to the Author:

The authors have very carefully addressed all my previous comments and the new data have further strengthened the manuscript. The RNA-seq data of the sorted NPCs clearly demonstrate that the timing of YY1-dependency is governed by the metabolic targets of YY1 that intriguingly matter less at later stages. I strongly support publishing this exciting manuscript as soon as possible.

Reviewer #2:

Remarks to the Author:

Thanks for the authors' work to make the data more solid, I have no more questions.

Reviewer #3:

Remarks to the Author:

The authors have adequately revised their manuscript.

Reviewer #4:

Remarks to the Author:

In this revised manuscript, Zurkirchen and colleagues made a good effort to address the concerns raised by the reviewers to strengthen their conclusion that YY1 regulates cerebral cortex development by acting in a stage-specific manner to regulate the proliferation and survival of neural progenitor cells (NPCs) and this requirement is due to a role in regulating a wide range of metabolic genes and protein synthesis. However, the fundamental flaw was not addressed as their observations could be equally due to flaws in their approach rather than real changes in stem cell behavior during development. The manuscript is, therefore, not of sufficient vigor and quality for publication in this journal.

Specifically, the key conclusion – that YY1 plays a stage-specific role in regulating brain development by showing that ablation of YY1 at different stages (e.g. E9.5 vs. E12.5) had different effect on NPCs – remains equally plausibly explained by alternative interpretation of the data presented.

As noted by the authors, Emx-IRES-Cre has been well characterized by others. The published studies, however, show this line is not specific to NPCs but also active in other cell types including neurons. Therefore, it is quite possible the effect on NPCs after ablation around E9.5 (cKO) was not due to a requirement for YY1 in NPCs themselves during early neurogenesis but secondary to defects elsewhere. Similarly, ablation of YY1 at E12.5 using CreERT2 (iKO) – showing a much milder effect – might simply have allowed the authors to bypass the effect on non-NPCs at earlier developmental stages. Moreover, it is also possible the knockout could bypass NPCs and affect neurons. Taking together with the time needed for tamoxifen-induce gene deletion to exhaust the YY1 proteins produced before gene deletion, it would not be surprising at all NPC phenotypes to be weaker using iKO. In other words, the conclusions by the authors could be due to comparing “apples” to “oranges” rather than NPCs at different stages of development.

The revised phrase –“The cell cycle length of apical RG cells decreases as the embryo develops” – is still incorrect as Arai et al (2011) were referring to different phases of the cell cycle.

REVIEWERS' COMMENTS:

Reviewer #1 (Remarks to the Author):

The authors have very carefully addressed all my previous comments and the new data have further strengthened the manuscript. The RNA-seq data of the sorted NPCs clearly demonstrate that the timing of YY1-dependency is governed by the metabolic targets of YY1 that intriguingly matter less at later stages. I strongly support publishing this exciting manuscript as soon as possible.

We are grateful for Reviewer #1's feedback and greatly appreciate the support to publish the manuscript. We are pleased to hear that we carefully addressed the Reviewer's comments.

Reviewer #2 (Remarks to the Author):

Thanks for the authors' work to make the data more solid, I have no more questions.

We would like to thank Reviewer #2 for the input and we are glad to hear that we successfully addressed his/her points.

Reviewer #3 (Remarks to the Author):

The authors have adequately revised their manuscript.

We would like to thank Reviewer #3 for the comments and we are glad to hear that we successfully revised the manuscript.

Reviewer #4 (Remarks to the Author):

In this revised manuscript, Zurkirchen and colleagues made a good effort to address the concerns raised by the reviewers to strengthen their conclusion that YY1 regulates cerebral cortex development by acting in a stage-specific manner to regulate the proliferation and survival of neural progenitor cells (NPCs) and this requirement is due to a role in regulating a wide range of metabolic genes and protein synthesis. However, the fundamental flaw was not addressed as their observations could be equally due to flaws in their approach rather than real changes in stem cell behavior during development. The manuscript is, therefore, not of sufficient vigor and quality for publication in this journal.

Specifically, the key conclusion – that YY1 plays a stage-specific role in regulating brain development by showing that ablation of YY1 at different stages (e.g. E9.5 vs. E12.5) had different effect on NPCs – remains equally plausibly explained by alternative interpretation of the data presented.

As noted by the authors, Emx-IRES-Cre has been well characterized by others. The published studies, however, show this line is not specific to NPCs but also active in other cell types including neurons. Therefore, it is quite possible the effect on NPCs after ablation around E9.5 (cKO) was not due to a requirement for YY1 in NPCs themselves during early neurogenesis but secondary to defects elsewhere. Similarly, ablation of YY1 at E12.5 using CreERT2 (iKO) – showing a much milder effect – might simply have allowed the authors to bypass the effect on non-NPCs at earlier developmental stages. Moreover, it is also possible

the knockout could bypass NPCs and affect neurons. Taking together with the time needed for tamoxifen-induced gene deletion to exhaust the YY1 proteins produced before gene deletion, it would not be surprising at all NPC phenotypes to be weaker using iKO. In other words, the conclusions by the authors could be due to comparing “apples” to “oranges” rather than NPCs at different stages of development.

The revised phrase –“The cell cycle length of apical RG cells decreases as the embryo develops” – is still incorrect as Arai et al (2011) were referring to different phases of the cell cycle.

As suggested by the Editor, we have added a paragraph on pp. 15/16 to clarify this issue. In particular, we now note that the observed phenotype appears to be due to a requirement for YY1 in NPCs rather than secondary to defects in neurons, given the very low number of neurons present at stages displaying the most prominent phenotype and, *vice versa*, the absence of a proliferation phenotype at later stages, when many neurons are present. In addition, siRNA-mediated knockdown experiments in isolated NPCs recapitulated the *Yy1cKO* phenotype, suggesting a cell-autonomous effect specifically in NPCs.

The sentence “The cell cycle length of apical RG cells decreases as the embryo develops” – was changed to: “Cell cycle length of NPCs generally decreases as the embryo develops.” (p. 19), as referenced in Calegari 2005, Takahashi 1995 and Caviness 1995.

We hope that with these changes we were able to address the remaining comments of Reviewer #4 and that, therefore, our study will now be acceptable for publication in *Nature Communications*.